# Variational Control for Guidance in Diffusion Models

**Kushagra Pandey** [* 1]  **Farrin Marouf Sofian** [* 1]  **Felix Draxler** [1]  **Theofanis Karaletsos** [2]  **Stephan Mandt** [1 3]

## Abstract

Diffusion models exhibit excellent sample quality, but existing guidance methods often require additional model training or are limited to specific tasks. We revisit guidance in diffusion models from the perspective of variational inference and control, introducing *Diffusion Trajectory Matching (DTM)* that enables guiding pretrained diffusion trajectories to satisfy a terminal cost. DTM unifies a broad class of guidance methods and enables novel instantiations. We introduce a new method within this framework that achieves state-of-the-art results on several linear, non-linear, and blind inverse problems without requiring additional model training or specificity to pixel or latent space diffusion models. Our code will be available at https://github.com/czi-ai/oc-guidance.

## 1. Introduction

Diffusion models (Sohl-Dickstein et al., 2015; Ho et al., 2020; Song et al., 2020) and related families (Lipman et al., 2023; Albergo & Vanden-Eijnden, 2023; Liu et al., 2023) exhibit excellent synthesis quality in large-scale generative modeling applications. Additionally, due to their principled design, these models exhibit great potential in serving as powerful generative priors for downstream tasks (Daras et al., 2024).

Consequently, guidance in diffusion models has received significant interest. However, the dominant approaches to classifier guidance (Dhariwal & Nichol, 2021) and classifier-free guidance (Ho & Salimans, 2021) require training additional models or retraining diffusion models for each conditioning task at hand, or are based on simplistic assumptions detrimental to sample quality (Kawar et al., 2022; Chung et al., 2022a; Song et al., 2023; Pandey et al., 2024b).

A trained diffusion model can be viewed as a steerable stochastic system that follows learned dynamics. By modifying its inputs, we can guide it in the spirit of stochastic optimal control — aiming to reach desired terminal states (e.g., matching a guidance signal) while staying close to its native trajectory distribution. This intuition motivates our formulation of classifier guidance as a *variational control* problem (Kappen, 2008). Inspired by *Control as Inference* frameworks (Kappen et al., 2012; Levine, 2018), we model guided diffusion as a controlled Markov process, where control signals are treated as variational parameters. We then apply variational inference to optimize these controls, ensuring that generated samples satisfy terminal constraints without straying far from the unconditional sample manifold (see Fig. 1a). We refer to this framework as *Diffusion Trajectory Matching* (DTM).

Recent work on steering diffusion models has already incorporated ideas from optimal control (Huang et al., 2024; Rout et al., 2025). However, these works focus on a restricted class of control problems. This obscures the available design choices revealed through our novel framework. Indeed, we find that DTM generalizes and explicitly contains a large class of prior work on guidance. We demonstrate the utility of this generalization by introducing a new sampling algorithm that seamlessly integrates with state-of-the-art diffusion model samplers like DDIM (Song et al., 2021a) and adapts well to diverse downstream tasks.

To summarize, we make the following contributions:

- We propose *Diffusion Trajectory Matching (DTM)*, a generalized framework for *training-free* guidance based on a variational control perspective. DTM subsumes many existing and novel guidance methods.

- We instantiate our framework as *Non-linear Diffusion Trajectory Matching (NDTM)*, which can be readily integrated with samplers like DDIM.

- We show that NDTM outperforms previous state-of-the-art baselines for solving challenging and diverse inverse problems with pretrained pixel-space (see Fig. 1) diffusion models. Furthermore, NDTM can be extended trivially to latent-space diffusion models like Stable Diffusion (Rombach et al., 2022) for tasks like style guidance (see Fig. 3).

---

[*]Equal contribution [1]Department of Computer Science, University of California, Irvine [2]Chan-Zuckerberg Initiative [3]Department of Statistics, University of California, Irvine. Correspondence to: Kushagra Pandey <pandeyk1@uci.edu>.

*Proceedings of the $42^{nd}$ International Conference on Machine Learning*, Vancouver, Canada. PMLR 267, 2025. Copyright 2025 by the author(s).

## 2. Background

**Diffusion Models.** Given a perturbation kernel $p(\mathbf{x}_t|\mathbf{x}_0) = \mathcal{N}(\mu_t\mathbf{x}_0, \sigma_t^2\mathbf{I}_d)$, diffusion models (Sohl-Dickstein et al., 2015; Ho et al., 2020) invert the noising process by learning a corresponding reverse process parameterized as,

$$\mathcal{Q}: q(\mathbf{x}_{0:T-1}|\mathbf{x}_T) = \prod_t q(\mathbf{x}_{t-1}|\mathbf{x}_t). \qquad (1)$$

The reverse diffusion posterior is specified as $q(\mathbf{x}_{t-1}|\mathbf{x}_t) = \mathcal{N}(\boldsymbol{\mu_\theta}(\mathbf{x}_t, t), \sigma_t^2\mathbf{I}_d)$ where $\boldsymbol{\mu_\theta}(.,.)$ is learned via score matching (Hyvärinen & Dayan, 2005; Vincent, 2011; Song & Ermon, 2019). Analogously continuous-time diffusion models (Song et al., 2020; Karras et al., 2022) assume that a *forward process*

$$d\mathbf{x}_t = f(t)\mathbf{x}_t\,dt + g(t)\,d\mathbf{w}_t, \quad t \in [0, T], \qquad (2)$$

with an drift $f(t)$ and diffusion coefficients $g(t)$ and standard Wiener process $\mathbf{w}_t$, converts data $\mathbf{x}_0 \in \mathbb{R}^d$ into noise $\mathbf{x}_T$. A *reverse* SDE specifies how data is generated from noise (Anderson, 1982; Song et al., 2020),

$$d\mathbf{x}_t = \left[f(t)\mathbf{x}_t - g(t)^2\nabla_{\mathbf{x}_t}\log p_t(\mathbf{x}_t)\right]\,dt + g(t)d\bar{\mathbf{w}}_t, \quad (3)$$

which involves the *score* $\nabla_{\mathbf{x}_t}\log p_t(\mathbf{x}_t)$ of the marginal distribution over $\mathbf{x}_t$ at time $t$. The score is intractable to compute and is approximated using a parametric estimator $\boldsymbol{s_\theta}(\mathbf{x}_t, t)$, trained using denoising score matching.

**Classifier Guidance in Diffusion Models.** Given a pretrained diffusion model $\boldsymbol{s_\theta}(\mathbf{x}_t, t)$, it is often desirable to guide the diffusion process conditioned on input $\boldsymbol{y}$. Consequently, the conditional diffusion dynamics read

$$d\mathbf{x}_t = \left[f(t)\mathbf{x}_t - g(t)^2\nabla_{\mathbf{x}_t}\log p(\mathbf{x}_t|\boldsymbol{y})\right]dt + g(t)d\bar{\mathbf{w}}_t. \quad (4)$$

In classifier guidance (Dhariwal & Nichol, 2021), the conditional score can be decomposed as

$$\nabla_{\mathbf{x}_t}\log p(\mathbf{x}_t|\boldsymbol{y}) = \boldsymbol{s_\theta}(\mathbf{x}_t, t) + \rho_t\nabla_{\mathbf{x}_t}\log p(\boldsymbol{y}|\mathbf{x}_t). \quad (5)$$

where $\rho_t$ is the guidance weight. The noisy likelihood score is often estimated by training a noise-conditioned estimator. It is also common to estimate this likelihood via $p(\boldsymbol{y}|\mathbf{x}_t) = \int p(\mathbf{x}_0|\mathbf{x}_t)p(\mathbf{y}|\mathbf{x}_0)d\mathbf{x}_0$. For example, Diffusion Posterior Sampling (DPS) (Chung et al., 2022a) approximates the diffusion posterior as, $p(\mathbf{x}_0|\mathbf{x}_t) = \delta(\mathbb{E}[\mathbf{x}_0|\mathbf{x}_t])$, where $\mathbb{E}[\mathbf{x}_0|\mathbf{x}_t]$ is Tweedie's estimate of the posterior at $\mathbf{x}_t$ (Efron, 2011). This approximation of the diffusion posterior in DPS results in a high sampling budget and high sensitivity to the gradient weight $\rho_t$. More expressive approximations (Song et al., 2023; Pandey et al., 2024b) result in specificity to linear inverse problems and thus cannot be extended to latent-space diffusion models trivially. We refer to Daras et al. (2024) for an in-depth discussion on explicit

approximations of the diffusion posterior. We will show in Section 3.4 that classifier guidance in diffusion models is a special case of our proposed framework. Next, we discuss our proposed framework, which generalizes to diverse tasks like inverse problems and style-guided generation without specificity to pixel or latent-space diffusion models.

## 3. Guidance with Diffusion Trajectory Matching (DTM)

We now propose a novel framework based on variational control for guidance in diffusion models. Our framework can be directly applied to pretrained diffusion models without requiring model retraining. For the remainder of our discussion, we restrict our attention to diffusion models and discuss an extension to flow-matching (Lipman et al., 2023) in Appendix A.3.

In the following, we first formulate guidance in discrete-time diffusion models as a variational optimal control problem (Section 3.1) following Kappen et al. (2012), which we refer to as *Diffusion Trajectory Matching* (DTM). We then present specific parameterizations of DTM in Section 3.2, which we work out as a guidance algorithm in Section 3.3. Lastly, in Appendix A.2, we transfer the DTM framework to continuous-time diffusion models (Song et al., 2020) and recover prior work in guidance in diffusion models.

### 3.1. Variational Control for Diffusion Guidance

The idea of our guidance framework is to take a controlled deviation from the *unguided diffusion trajectory* implied by Eq. (1) in Section 2, which we repeat for convenience:

$$\mathcal{Q}: q(\mathbf{x}_{0:T-1}|\mathbf{x}_T) = \prod_t q(\mathbf{x}_{t-1}|\mathbf{x}_t). \qquad (6)$$

To steer the trajectory towards a target state fulfilling external constraints, we introduce a control signal $\mathbf{u}_t$ at every time $t$. This yields the following *guided* dynamics for a given initial state $\mathbf{x}_T$:

$$\mathcal{P}: p(\mathbf{x}_{0:T-1}|\mathbf{x}_T, \mathbf{u}_{1:T}) = \prod_t p(\mathbf{x}_{t-1}|\mathbf{x}_t, \mathbf{u}_t). \quad (7)$$

While we model the guided dynamics as Markovian due to convenience, non-Markovian approximations are also possible (Li et al., 2021). Given a set of external constraints, the task is to determine the variational control $\mathbf{u}_t$. Consequently, following Kappen et al. (2012), we can pose this problem as a stochastic optimal control problem with the terminal and transient costs formulated as,

$$\mathcal{C}(\mathbf{x}_T, \mathbf{u}_{1:T}) = w_T\underbrace{\mathbb{E}_{\mathbf{x}_0\sim p}[\Phi(\mathbf{x}_0)]}_{\text{Terminal Cost }\mathcal{C}_{\text{te}}} + \underbrace{D_{\text{KL}}(\mathcal{P} \parallel \mathcal{Q})}_{\text{Transient Cost }\mathcal{C}_{\text{tr}}}. \quad (8)$$

The terminal cost in Eq. (8) encodes desirable constraints on the final guided state while the transient cost ensures

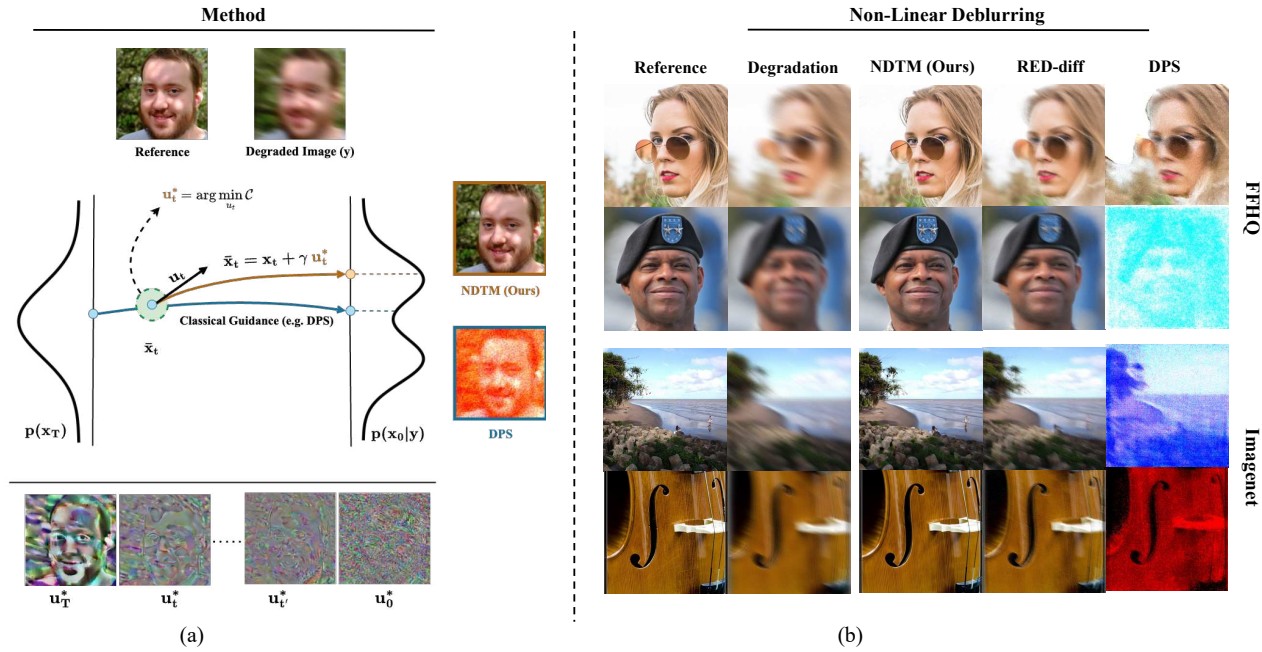

*Figure 1.* **Our method guides diffusion sampling to fulfill external constraints.** To this end, we optimize the local direction $\mathbf{u}_t^*$ via external constraints while respecting the original trajectory, see Eq. (11) (left, center). This recovers more accurate reconstructions across tasks compared to classical guidance methods: Nonlinear deblurring (Right). Our method accurately captures most details, while competing methods introduce artifacts in the generated reconstructions.

that the guided trajectory does not deviate strongly from the unguided trajectory, so that the final guided state $\mathbf{x}_0$ lies near the image manifold. The two losses are traded by a scalar $w_T$.

**Choice of Terminal Cost.** The terminal cost in Eq. (8) encodes desirable constraints on the final guided state. For instance, $\Phi(\mathbf{x}_0) \propto -\log p(\boldsymbol{y}|\mathbf{x}_0)$ could be the log-likelihood of a probabilistic classifier for class-conditional generation or the degradation process for solving inverse problems (see Section 5 for exact form of terminal costs for different tasks). For instance, Huang et al. (2024) adapts guidance for a non-differentiable terminal cost using a path integral control (Kappen, 2005) formulation. While an interesting direction for further work, we only assume that the terminal cost is differentiable for now.

**Choice of Divergence.** We use the KL-Divergence as it decomposes over individual timesteps,

$$\mathcal{C}_{\text{tr}} = \sum_t \mathbb{E}_{\mathbf{x}_t}\left[ D_{\text{KL}}(p(\mathbf{x}_{t-1}|\mathbf{x}_t, \mathbf{u}_t) \,\|\, q(\mathbf{x}_{t-1}|\mathbf{x}_t)) \right]. \quad (9)$$

Note that other divergence measures can be useful depending on the specific form of the diffusion posterior (Nachmani et al., 2021; Zhou et al., 2023; Pandey et al., 2025; Holderrieth et al., 2025).

**Simplifications.** The proposed loss in Eq. (8) is generic and

principled, but is difficult to jointly optimize for all controls $\mathbf{u}_{1:T}$ due to the need to backpropagate through the entire diffusion trajectory. To avoid this computational overhead, we make several simplifications that make the objective computationally tractable. We justify the validity of the modifications through our empirical results in Section 5.

First, we optimize $\mathbf{u}_t$ in a greedy manner, that is at any time $t$ in the diffusion process we optimize $\mathbf{u}_t$, assuming that the remaining steps $t - 1, \ldots, 1$ are unguided. After optimizing for $\mathbf{u}_t$, we sample from the optimized posterior $\mathbf{x}_{t-1} \sim p(\mathbf{x}_{t-1}|\mathbf{x}_t, \mathbf{u}_t^*)$ and iterate. When the variational control of $\mathbf{u}_t$ is flexible enough, suboptimal greedy choices early in the trajectory can be compensated for later.

Second, we evaluate the terminal cost at the current *expected* final guided state via Tweedie's Formula for $\mathbb{E}[\mathbf{x}_0|\mathbf{x}_t, \mathbf{u}_t]$:

$$\mathcal{C}_{\text{te}} = \mathbb{E}_{\mathbf{x}_0}[\Phi(\mathbf{x}_0)] \approx \Phi(\mathbb{E}[\mathbf{x}_0|\mathbf{x}_t, \mathbf{u}_t]) = \Phi(\hat{\mathbf{x}}_0^t) \quad (10)$$

where we have approximated $p(\mathbf{x}_0|\mathbf{x}_t, \mathbf{u}_t) \approx \delta(\mathbf{x}_0 - \hat{\mathbf{x}}_0^t)$.

**Diffusion Trajectory Matching (DTM).** Together, the optimization problem to solve at time $t$ given a position $\mathbf{x}_t$ reads:

$$\mathcal{C}(\mathbf{u}_t) = w_T\Phi(\hat{\mathbf{x}}_0^t) + D_{\text{KL}}(p(\mathbf{x}_{t-1}|\mathbf{x}_t, \mathbf{u}_t) \,\|\, q(\mathbf{x}_{t-1}|\mathbf{x}_t)). \quad (11)$$

We refer to Eq. (11) as *Diffusion Trajectory Matching* (DTM).

**Algorithm 1** NDTM (DDIM). Sampling proceeds by inferring the control (shaded) followed by sampling from the guided posterior (shaded) at any time $t$

1: **Input:** Optimization Steps: N, Guidance: $\gamma$, Pretrained denoiser: $\epsilon_{\boldsymbol{\theta}}(.,.)$, Timestep schedule: $\{t\}_{j=0}^{T}$, DDIM Coefficients: $\alpha_t, \sigma_t$, Loss Weights: $\tau_t, \kappa_t, w_T$
2: **Initialization:** $\mathbf{x}_T \sim \mathcal{N}(0, \boldsymbol{I}_d)$
3: **for** $t = T$ **to** $1$ **do**
4:     $\mathbf{u}_t^{(0)} = 0$
5:     $\hat{e}_{\text{uncond}} \leftarrow \epsilon_\theta(\mathbf{x}_t, t)$

6:     **for** $i = 0$ **to** $N - 1$ **do**
7:         $\hat{e}_{\text{control}}^{(i)} \leftarrow \epsilon_\theta(\mathbf{x}_t + \gamma \boldsymbol{u}_t^{(i)}, t)$
8:         $\hat{\mathbf{x}}_0^{(i)} \leftarrow \mathbb{E}[\mathbf{x}_0 | \mathbf{x}_t + \gamma \boldsymbol{u}_t^{(i)}]$
9:         $\mathcal{C}_{\text{score}} = \tau_t^2 \left\| \hat{e}_{\text{uncond}} - \hat{e}_{\text{control}}^{(i)} \right\|_2^2$
10:        $\mathcal{C}_{\text{control}} = \kappa_t^2 \|\mathbf{u}_t\|_2^2$
11:        $\mathcal{C}_{\text{terminal}} = w_T \Phi(\hat{\mathbf{x}}_0^{(i)})$
12:        $\mathcal{C}_t^{(i)} \leftarrow \mathcal{C}_{\text{score}} + \mathcal{C}_{\text{control}} + \mathcal{C}_{\text{terminal}}$
13:        $\boldsymbol{u}_t^{(i+1)} \leftarrow \texttt{Update}(\boldsymbol{u}_t^{(i)}, \nabla_{u_t} \mathcal{C}_t^{(i)})$
14:     **end for**

15:     $\mathbf{x}_{t-1} \leftarrow \text{DDIM}(\mathbf{x}_t + \gamma \mathbf{u}_t^*, t)$

16: **end for**
17: return $\mathbf{x}_0$

**Continuous-Time Variants.** To apply DTM to continuous-time diffusion and flow matching, we adapt the transient costs $\mathcal{C}_{\text{tr}}$. We call the following *Continuous-Time Diffusion Trajectory Matching*, derived for continuous-time diffusion (Song et al., 2020) in Appendix A.2:

$$\mathcal{C}_{\text{tr}} = \frac{g(t)^2}{2} \mathbb{E}_{\mathbf{x}_t} \left[ \left\| s_{\boldsymbol{\theta}}(\mathbf{x}_t, t) - s_{\boldsymbol{\theta}}(\mathbf{x}_t, \mathbf{u}_t, t) \right\|_2^2 \right]. \quad (12)$$

Similarly, for flow matching (Lipman et al., 2023; Liu et al., 2023; Albergo et al., 2023), we refer to this as *Flow Trajectory Matching* (FTM), see Appendix A.3:

$$\mathcal{C}_{\text{tr}} = \| v_{\boldsymbol{\theta}}(\mathbf{x}_t, t) - v_{\boldsymbol{\theta}}(\mathbf{x}_t, \mathbf{u}_t, t) \|^2. \quad (13)$$

Next, we present a specific parameterization of our framework and its instantiation using standard diffusion models.

### 3.2. Non-linear Diffusion Trajectory Matching (NDTM)

In the context of Gaussian diffusion models (Ho et al., 2020; Song et al., 2020; Karras et al., 2022), the unguided diffusion posterior is often parameterized as,

$$q(\mathbf{x}_{t-1} | \mathbf{x}_t) = \mathcal{N}(\boldsymbol{\mu}_\theta(\mathbf{x}_t, t), \sigma_t^2 \boldsymbol{I}_d) \quad (14)$$

In analogy to how the unguided diffusion denoising process $q$ is parameterized, we define our controlled process $p$ as

$$p(\mathbf{x}_{t-1} | \mathbf{x}_t, \mathbf{u}_t) = \mathcal{N}(\boldsymbol{\mu}_\theta(\mathbf{x}_t, \mathbf{u}_t, t), \sigma_t^2 \boldsymbol{I}_d). \quad (15)$$

From a practical standpoint, since unconditional score models are usually parameterized using neural networks with an input noisy state and a timestep embedding, we further parameterize the posterior mean $\boldsymbol{\mu}_\theta(\mathbf{x}_t, \mathbf{u}_t, t) = \boldsymbol{\mu}_\theta(\boldsymbol{f}(\mathbf{x}_t, \mathbf{u}_t, t), t)$, where the *aggregation* function $\boldsymbol{f} : \mathbb{R}^d \times \mathbb{R}^d \times \mathbb{R} \to \mathbb{R}^d$ combines the noisy state $\mathbf{x}_t$ and the control $\mathbf{u}_t$ appropriately. In this work, we choose an additive form of $\boldsymbol{f} = \mathbf{x}_t + \gamma \mathbf{u}_t$ where $\gamma$ is the *guidance weight* used to update the current noisy state $\mathbf{x}_t$ in the direction of the control signal $\mathbf{u}_t$. We leave exploring other aggregation functions as future work. Moreover, in practice, we sample from a single diffusion trajectory and therefore omit the expectation in Eq. (11). Consequently, the control cost in Eq. (11) can be simplified as,

$$\mathcal{C}(\mathbf{u}_t) = \|\boldsymbol{\mu}_{\boldsymbol{\theta}}(\mathbf{x}_t + \gamma \mathbf{u}_t, t) - \boldsymbol{\mu}_{\boldsymbol{\theta}}(\mathbf{x}_t, t)\|_2^2 + w_T \Phi(\hat{\mathbf{x}}_0^t). \quad (16)$$

Due to the non-linear dependence of the guided posterior on the control signal $\mathbf{u}_t$, we refer to the transient cost specification in Eq. 16 as *Non-Linear Diffusion Trajectory Matching* (NDTM). We will show in Section 3.4 that linear control can be formulated as a special case of this parameterization, yielding classifier guidance. Next, we instantiate the NDTM objective practically.

### 3.3. Specific Instantiations

Here, we present a simplified form of the NDTM objective in the context of DDIM (Song et al., 2021a).

**Proposition 3.1.** *For the diffusion posterior parameterization in DDIM (Song et al., 2021a), the NDTM objective in Eq. 16 has the following tractable upper bound (see proof in Appendix A.1),*

$$\mathcal{C}(\mathbf{u}_t) \leq \kappa_t^2 \|\mathbf{u}_t\|_2^2 + \tau_t^2 \|\epsilon_\theta(\bar{\mathbf{x}}_t, t) - \epsilon_\theta(\mathbf{x}_t, t)\|_2^2 + w_T \Phi(\hat{\mathbf{x}}_0^t), \quad (17)$$

*where $\bar{\mathbf{x}}_t = \mathbf{x}_t + \gamma \mathbf{u}_t$ is the guided state and the coefficients $\kappa_t = \frac{\gamma \sqrt{\alpha_{t-1}}}{\sqrt{\alpha_t}}$ and $\tau_t = \sqrt{1 - \alpha_{t-1} - \sigma_t^2} - \frac{\sqrt{\alpha_{t-1}(1-\alpha_t)}}{\sqrt{\alpha_t}}$ are time-dependent scalars.*

The coefficients $\alpha_t$ and $\sigma_t$ are specific to DDIM (see Appendix A.1 for more details). Intuitively, the simplified NDTM loss in Eq. (17) measures the deviation between the guided and unguided dynamics, penalizing the magnitude of the control signal $\mathbf{u}_t$ (first term) and deviations in the noise predictions (second term). On the contrary, the terminal loss ensures that the *expected* final guided state satisfies the external constraints. Therefore, the first two terms in Eq. (17) act as regularizers on the control signal $\mathbf{u}_t$. In Appendix A.2, we derive this simplification also for continuous-time diffusion models. Lastly, it is worth noting that the control loss in Eq. 17 can be generalized as,

$$\mathcal{C}(\mathbf{u}_t) = w_c \|\mathbf{u}_t\|_2^2 + w_s \|\epsilon_\theta(\bar{\mathbf{x}}_t, t) - \epsilon_\theta(\mathbf{x}_t, t)\|_2^2 + w_T \Phi(\hat{\mathbf{x}}_0^t) \quad (18)$$

In this work, unless specified otherwise, we set $w_s = \tau_t^2$, $w_c = \kappa_t^2$. However, exploring alternative weighting schedules for different loss terms can be an interesting direction for further work.

**Putting it all together.** To summarize, at each diffusion time step $t$, we estimate the control signal $\mathbf{u}_t$ by minimizing the cost $\mathcal{C}(\mathbf{u}_t)$ (for instance Eq. (17) for DDIM). This iterative optimization allows the model to dynamically adjust the control to best align the trajectory with the desired terminal cost while minimizing deviations with the unguided diffusion trajectory. Finally, we sample from the guided posterior $p(\mathbf{x}_{t-1}|\mathbf{x}_t, \mathbf{u}_t^*)$. We provide a visual illustration of the NDTM algorithm in Fig. 1a and its pseudocode implementation in Algorithm 1.

### 3.4. Connection to Existing Guidance Mechanisms

In this section, we rigorously establish a connection between optimal control and classifier guidance: Our variational formulation in Section 3.1 captures existing approaches. We derive this result in the continuous-time variant, as this allows for a closed-form solution of the control problem.

In particular, let us choose a linear parameterization of the guided score in Eq. (12), that is $s_\theta(\mathbf{x}_t, \mathbf{u}_t, t) = s_\theta(\mathbf{x}_t, t) + \mathbf{u}_t$. Then, the transient cost reduces to:

$$\mathcal{C}_{\text{tr}} = \int \|\mathbf{u}_t\|^2 dt. \tag{19}$$

This is exactly the case of the well-established *Path Integral Control* (Kappen, 2005; 2008). The solution of this optimal control problem in Eq. (8) reads (Kappen, 2008, Eq. (34)):

$$\mathbf{u}_t^* = g(t)w_T \nabla_{\mathbf{x}_t} \log \mathbb{E}_{p(\mathbf{x}_0|\mathbf{x}_t)}[\exp(-\Phi(\mathbf{x}_0))]. \tag{20}$$

Notably, if the terminal cost takes the form of a classifier likelihood $\Phi(\mathbf{x}_0) \propto -\log p(\mathbf{y}|\mathbf{x}_0)$, it can be shown (Huang et al., 2024) that the optimal control simplifies to classifier guidance (Dhariwal & Nichol, 2021): $\mathbf{u}_t^* = g(t)w_T \nabla_{\mathbf{x}_t} p(\mathbf{y}|\mathbf{x}_t)$. This puts a large class of methods approximating the expectation over the posterior $p(\mathbf{x}_0|\mathbf{x}_t)$ (Chung et al., 2022a; Song et al., 2023; Pandey et al., 2024b; Huang et al., 2024) into perspective: In terms of our DTM framework, they perform optimal control with a linear control mechanism (i.e. control added linearly to the score function). Empirically, we will see in Section 5 that our generalization to non-linear control provides significant performance improvements.

## 4. Related Work

**Conditional Diffusion Models.** In general, the conditional score $\nabla_{\mathbf{x}_t} \log p(\mathbf{x}_t|\boldsymbol{y})$ needed for guided sampling can be learned during training (Saharia et al., 2022; Podell et al., 2024; Rombach et al., 2022) or approximated during inference (see Daras et al. (2024) for a detailed review).

Here, we focus on training-free guidance during inference. In this context, there has been some recent progress in approximating the noisy likelihood score (see Eq. 5) by approximating the diffusion posterior $p(\mathbf{x}_0|\mathbf{x}_t)$. For instance, DPS (Chung et al., 2022a) approximates the diffusion posterior by a Dirac distribution centered on Tweedie's estimate (Efron, 2011). This has the advantage that the guidance can be adapted to linear and non-linear tasks alike. However, due to a crude approximation, DPS converges very slowly and, in our observation, could be unstable for certain tasks (see Table 1). Consequently, some recent work (Yu et al., 2023; Bansal et al., 2024) adds a correction term on top of the DPS update rule to better satisfy the constraints. MPGD (He et al., 2024) attempts to alleviate some of these issues by leveraging the manifold hypothesis. In contrast, our proposed method instead directly estimates the guided posterior at each sampling step, thus sidestepping the limitations of DPS in the first place.

It is worth noting that our method resembles DCPS (Janati et al., 2024), which adopts a similar approach of learning a series of potentials to sample from the final posterior. However, DCPS involves an additional overhead of Langevin Monte Carlo sampling in addition to the posterior optimization step. Moreover, the control perspective adopted in this work helps contextualize prior work in guidance within our framework (see Section 3.4), which is lacking in DCPS. Some recent methods like TFG (Ye et al., 2024) also propose general frameworks for guidance in diffusion models. However, our proposed method does not fit within their framework.

More recent work (Kawar et al., 2022; Wang et al., 2023; Song et al., 2023; Pandey et al., 2024b; Pokle et al., 2024; Boys et al., 2024) relies on expressive approximations of the diffusion posterior. While this can result in accurate guidance and faster sampling, a large proportion of these methods are limited to linear inverse problems, which further limits their application to pixel space diffusion models. In contrast, our method can be adapted to generic inverse problems and is thus agnostic to the diffusion model architecture. Lastly, another line of work in inverse problems approximates the data posterior $p(\mathbf{x}_0|\mathbf{y})$ using variational inference (Blei et al., 2017; Zhang et al., 2018a). For instance, RED-diff (Mardani et al., 2024) proposes to learn an unimodal approximation to the data posterior by leveraging a diffusion prior. However, this can be too restrictive in practice and comes at the expense of blurry samples. We refer interested readers to Daras et al. (2024) for a more detailed review of training-free methods for solving inverse problems in diffusion models.

**Optimal Control for Diffusion Models.** There has been some recent interest in exploring connections between stochastic optimal control and diffusion models (Berner

et al., 2024). Chen et al. (2024) leverages ideas from control theory for designing efficient diffusion models with straight-line trajectories in augmented spaces (Pandey & Mandt, 2023). Since our guided sampler can be used with any pre-trained diffusion models, our approach is complementary to this line of work. More recently, SCG (Huang et al., 2024) leverages ideas from path integral control to design guidance schemes with non-differentiable constraints. In contrast, we only focus on differentiable terminal costs, and extending our framework to non-differentiable costs could be an important direction for future work. Lastly, Rout et al. (2025) propose RB-Modulation, a method based on control theory for personalization using diffusion models. Interestingly, while RB-Modulation is primarily inspired by a class of tractable problems in control theory, it is a special case of our framework in the limit of $w_s = 0, w_c = 0$ and $\gamma = 1$. Therefore, our proposed framework is more flexible.

# 5. Experiments

While our method serves as a general framework for guidance in diffusion models, here, we focus on solving inverse problems and style-guided generation. Through both quantitative and qualitative results, we demonstrate that our approach outperforms recent state-of-the-art baselines across these tasks using pretrained diffusion models. Lastly, we emphasize key design parameters of our proposed method as ablations. We defer all implementation details to App. B.

**Problem Setup and Terminal Costs:** For inverse problems, given a corruption model $\mathcal{A}$ and a noisy measurement $\mathbf{y} \in R^d$ the goal is to recover the unknown sample $\mathbf{x}_0 \sim p_{\text{data}}$, from the degradation $\mathbf{y} = \mathcal{A}(\boldsymbol{x}_0) + \sigma_y \mathbf{z}, \quad \mathbf{z} \sim \mathcal{N}(\mathbf{0}, \boldsymbol{I}_d)$. For linear inverse problems, $\mathbf{y} = \mathcal{A}\boldsymbol{x}_0$. In the case where only the functional form of the degradation operator $\mathcal{A}$ is known but its parameters are not, the problem is known as *Blind Inverse Problem*. For inverse problems, we consider the following form of the terminal cost $\Phi(\hat{\mathbf{x}}_0^t)$ in Eq. 17,

$$\Phi(\hat{\mathbf{x}}_0^t) = \|\boldsymbol{y} - \mathcal{A}(\hat{\mathbf{x}}_0^t)\|_2^2 \tag{21}$$

where $\hat{\mathbf{x}}_0^t$ is the Tweedies estimate at any given time t. We also consider the task of style guidance with Stable Diffusion, where the goal is to generate samples adhering to a specific prompt and a reference *style* image. More specifically, given a reference style image $\boldsymbol{r}$, a pretrained feature extractor (like CLIP) $\mathcal{F}$, a pretrained decoder $\mathcal{D}$ such that $\hat{\mathbf{x}}_0^t = \mathcal{D}(\hat{\mathbf{z}}_0^t)$, we define $\Phi(\hat{\mathbf{x}}_0^t)$ as,

$$\Phi(\hat{\mathbf{x}}_0^t) = \|\mathcal{G}(\mathcal{F}(\boldsymbol{r})) - \mathcal{G}(\mathcal{F}(\hat{\mathbf{x}}_0^t))\|_F^2 \tag{22}$$

where $\mathcal{G}$ denotes the Gram-matrix operation and $\|.\|_F$ denotes the Frobenius norm. Note that prompt adherence is achieved via the pretrained Stable Diffusion model.

**Models and Datasets:** For inverse problems, we conduct experiments on the FFHQ $(256 \times 256)$ (Karras et al., 2019)

*Table 1.* Comparisons on **noisy Non-linear Deblur**. NDTM outperforms competing baselines by a significant margin. **Bold**: best.

| Method | FFHQ ($256 \times 256$) | | | ImageNet ($256 \times 256$) | | |
|---|---|---|---|---|---|---|
| | LPIPS↓ | FID↓ | KID↓ | LPIPS↓ | FID↓ | KID↓ |
| DPS | 0.752 | 249.01 | 0.139 | 0.888 | 346.82 | 0.2186 |
| RED-diff | 0.362 | 64.57 | 0.036 | 0.416 | 78.07 | 0.0224 |
| MPGD | 0.636 | 113.98 | 0.086 | 0.832 | 148.96 | 0.085 |
| RB-Modulation | 0.064 | 19.92 | 0.0032 | 0.249 | 47.60 | 0.0078 |
| NDTM (ours) | **0.046** | **14.198** | **0.0004** | **0.163** | **34.31** | **0.0032** |

and ImageNet $(256 \times 256)$ (Deng et al., 2009) datasets, using a held-out validation set of $1,000$ samples from each. For FFHQ, we use the pre-trained model provided by Chung et al. (2022a), and for ImageNet, we use the unconditional pre-trained checkpoint from OpenAI (Dhariwal & Nichol, 2021). For style guidance, following MPGD (He et al., 2024), we randomly generate 1k (prompt, image) pairs using images from WikiArt (Saleh & Elgammal, 2015) and prompts from PartiPrompt (Yu et al., 2022). We use the pre-trained CLIP encoder and Stable Diffusion 1.4 models as the feature extractor $\mathcal{F}$ and diffusion model, respectively.

**Tasks and Metrics:** For inverse problems, we consider random inpainting, super-resolution, and non-linear deblurring. Additionally, we consider blind image deblurring (BID) task where we additionally infer the deblurring kernel $\boldsymbol{k}$ along with the final reconstruction. We set the noise level $\sigma_y = 0.01$ for all inverse problems.

For quantitative evaluation on inverse problems, we report metrics optimized for perceptual quality, including Learned Perceptual Image Patch Similarity (LPIPS) (Zhang et al., 2018b), Fréchet Inception Distance (FID) (Heusel et al., 2017), and Kernel Inception Distance (KID) (Bińkowski et al., 2018). For completeness, recovery metrics like the Peak Signal-to-Noise Ratio (PSNR) are provided in Appendix C. With the exception of BID (for which we use 100 images), we evaluate all other inverse problems on 1k images. For style guidance, following prior work Yu et al. (2023); He et al. (2024), we report the CLIP score (which measures prompt adherence) and the Style Score (which measures style adherence) on 1k text and image pairs.

## 5.1. Inverse Problems

We first evaluate the proposed NDTM sampler against competing baselines for non-linear, blind, and linear inverse problems. We provide all hyperparameter details for our method and competing baselines in Appendix B.

**Non-Linear Deblurring.** We consider non-linear deblurring with the same setup as in Chung et al. (2022a). For this task, we compare against DPS (Chung et al., 2022a), RED-Diff (Mardani et al., 2024), MPGD (He et al., 2024),

*Table 2.* NDTM performs on-par/better than competing baselines on noisy linear inverse problems. Missing entries indicate unstable performance after multiple tuning attempts **Bold**: best.

| | Super-Resolution (4x) | | | | | | Random Inpainting (90%) | | | | | |
| --- | --- | --- | --- | --- | --- | --- | --- | --- | --- | --- | --- | --- |
| | FFHQ (256 × 256) | | | Imagenet (256 × 256) | | | FFHQ (256 × 256) | | | Imagenet (256 × 256) | | |
| Method | LPIPS↓ | FID↓ | KID↓ | LPIPS↓ | FID↓ | KID↓ | LPIPS↓ | FID↓ | KID↓ | LPIPS↓ | FID↓ | KID↓ |
| DPS | 0.061 | 20.61 | 0.0029 | 0.195 | 30.67 | 0.0021 | **0.058** | 20.24 | **0.0019** | 0.152 | 32.56 | 0.0023 |
| DDRM | 0.116 | 36.13 | 0.0183 | 0.325 | 52.76 | 0.0151 | 0.582 | 167.57 | 0.1530 | 0.791 | 211.66 | 0.1517 |
| RED-diff | 0.151 | 41.54 | 0.0179 | 0.354 | 51.83 | 0.0084 | 0.430 | 155.49 | 0.1370 | 0.633 | 218.88 | 0.1531 |
| MPGD | 0.119 | 28.54 | 0.0032 | 0.215 | 37.39 | 0.0017 | 0.658 | 173.28 | 0.134 | 0.908 | 156.44 | 0.053 |
| C-ΠGDM | 0.106 | 29.61 | 0.0073 | 0.270 | 39.96 | 0.0024 | 0.551 | 137.85 | 0.1020 | - | - | - |
| RB-Modulation | **0.054** | **18.22** | **0.0013** | 0.211 | 35.26 | 0.0032 | 0.091 | 25.76 | 0.0026 | 0.223 | 46.30 | 0.0066 |
| NDTM (ours) | **0.054** | 18.99 | 0.0019 | **0.158** | **28.75** | **0.0011** | 0.059 | **20.11** | 0.0020 | **0.149** | **30.43** | **0.0018** |

*Table 3.* Comparisons on **noisy Blind Image Deblurring (BID)** for the FFHQ 256×256 dataset. NDTM outperforms DMPlug (Wang et al., 2024) by a significant margin while requiring an order of magnitude less sampling time (reported in minutes/img). **Bold**: best. †: N=15, T=200, ζ: N=15, T=400.

| | Gaussian blur | | | Motion blur | | |
| --- | --- | --- | --- | --- | --- | --- |
| Method | LPIPS↓ | FID↓ | Time↓ | LPIPS↓ | FID↓ | Time↓ |
| DMPlug | 0.147 | 69.36 | 51.24 | 0.118 | 72.85 | 51.13 |
| NDTM† (Ours) | 0.103 | 55.15 | **7.17** | 0.086 | 49.99 | **7.17** |
| NDTMζ (Ours) | **0.083** | **47.34** | 18.07 | **0.063** | **38.6** | 18.13 |

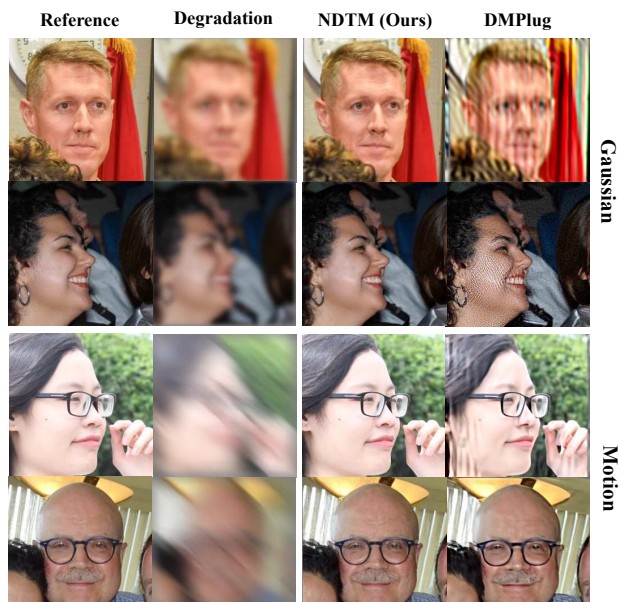

*Figure 2.* **NDTM outperforms competing baselines on blind image deblurring (BID) with Gaussian (top) and Motion (bottom) kernels**. NDTM accurately captures most details, while competing methods introduce artifacts in the generated reconstructions.

and RB-Modulation (Rout et al., 2025) (which is a special case of our method with $\gamma_t = 1.0$, $w_s = 0$, $w_c = 0$). Figure 1 (Right) illustrates the comparison between competing baselines and our proposed method, NDTM, on this task. Qualitatively, we find that DPS is very sensitive to guidance step size and is usually unstable on this task. Moreover, while RED-diff does not have stability issues, it is biased towards generating blurry samples. This is not surprising given their unimodal approximation to the data posterior $p(\mathbf{x}_0|\mathbf{y})$. On the contrary, NDTM generates high-fidelity reconstructions with a stable sampling process. Similarly, our quantitative results in Table 1 validate our qualitative findings as our method outperforms competing baselines on perceptual quality for both datasets by a significant margin.

**Blind Image Deblurring (BID).** Next, we extend our framework to blind image deblurring, maintaining the same setup as DMPlug (Wang et al., 2024). We compare against DM-Plug, which, to the best of our knowledge, is also the state-of-the-art method for this task. Interestingly, adapting our proposed method for blind inverse problems only involves jointly optimizing the unknown blur kernel parameters along with the control $\mathbf{u}_t$. More specifically, for degradation of the form $\mathbf{y} = \mathbf{k} * \mathbf{x}_0 + \sigma_y \mathbf{z}$ with unknown blurring kernel $\mathbf{k}$, we update line 11 in Algorithm 1 as $\mathcal{C}_{\text{terminal}} = w_T \Phi(\hat{\mathbf{x}}_0^{(i)}, k^{(i)})$ and optimizing for the trainable kernel for each image, as $\mathbf{k}^{(i+1)} \leftarrow \text{Update}(\mathbf{k}^{(i)}, \nabla_k \mathcal{C}_t^{(i)})$.

Figure 2 (Right) illustrates the comparison between DM-Plug and NDTM adapted for this task. Qualitatively, we find that while DMPlug can introduce artifacts in generating reconstructions, NDTM generates high-quality reconstructions. Table 3 further validates our qualitative findings as our method outperforms DMPlug on perceptual quality metrics. More interestingly, while DMPlug is extremely expensive for a single image, our method outperforms the former on sample quality by a significant margin while being an order of magnitude faster (see Table 3). This illustrates that our sampler has a more efficient way to trade sampling speed for quality. We present additional results in Appendix C.

**Linear Inverse Problems.** Lastly, we compare competing

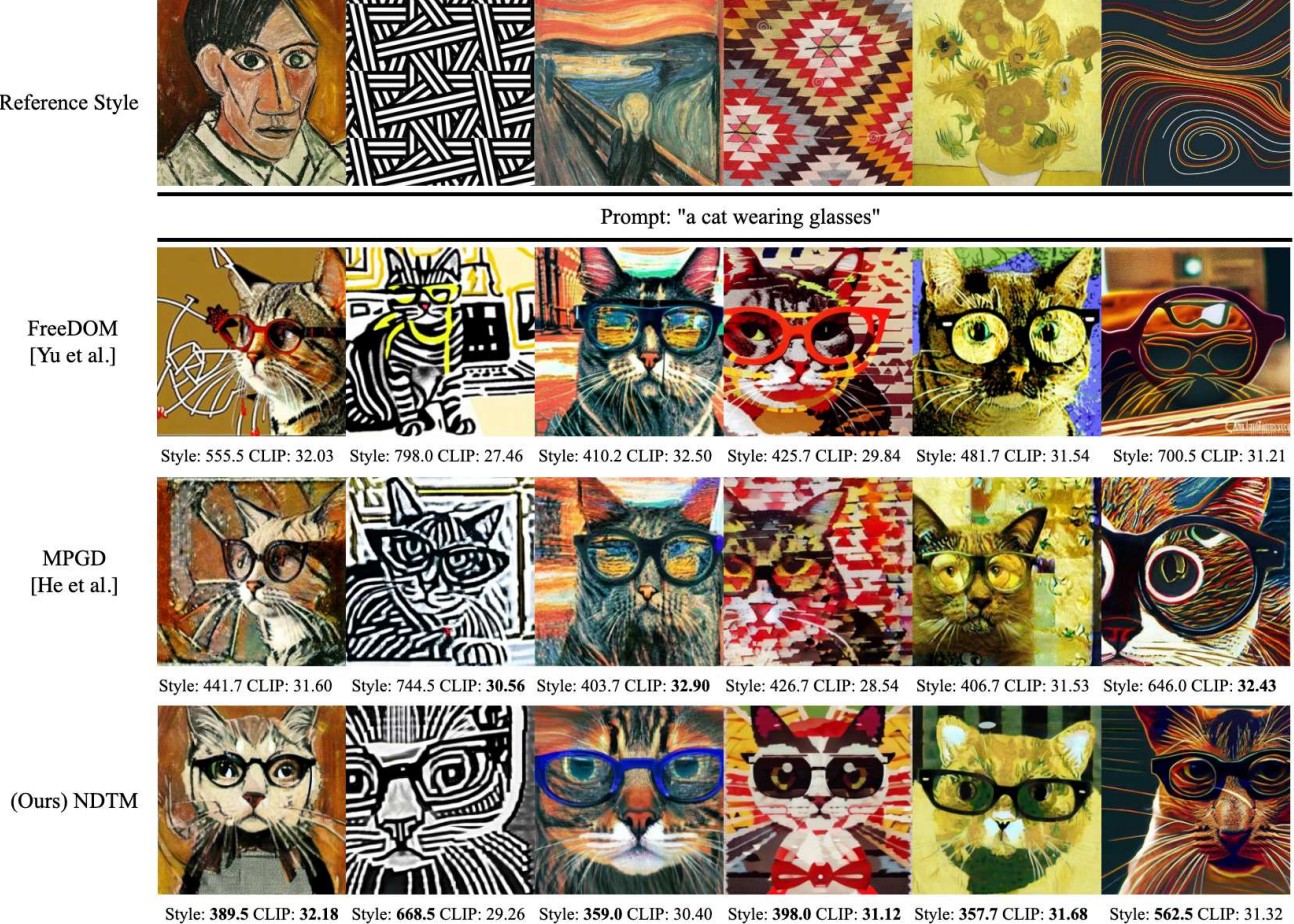

*Figure 3.* **Style Guidance with Stable Diffusion.** NDTM (proposed) provides a better tradeoff between Prompt adherence and Style adherence using Stable Diffusion 1.4. While baselines can introduce random artifacts in generated samples, NDTM preserves sample quality while exhibiting better style adherence. **(Top Panel)** Reference Style Images. **(Bottom Panel)** Samples corresponding to the reference style images in the Top Panel. The CLIP (Radford et al., 2021) score (higher is better) represents the similarity between the generated image and the text prompt, indicating prompt adherence. The Style score (lower is better) represents the distance between CLIP features for the reference style and the generated image, indicating style adherence. We present additional quantitative results in Table 4.

methods on linear inverse problems: (4x) Super-resolution and Random inpainting with a 90% masking probability. In addition to the baselines used for the non-linear deblur task, we also compare against DDRM (Kawar et al., 2022) and C-ΠGDM (Pandey et al., 2024b). As illustrated in Table 2, for super-resolution, NDTM outperforms competing baselines for both datasets. For random inpainting, our method performs comparably with DPS on the FFHQ dataset. However, for a more difficult benchmark like ImageNet, NDTM outperforms the next best competing baseline, DPS, on this task. We present additional qualitative results for linear inverse problems in Appendix C

### 5.2. Style Guidance

The goal in style guidance is to generate samples that adhere to a specified text prompt and the style of a reference

|  | CLIP Score ↑ | Style Score ↓ |
|---|---|---|
| FreeDOM | 30.86 | 508.28 |
| MPGD | 30.21 | 498.85 |
| (Ours) NDTM | **31.34** | **475.62** |

*Table 4.* Quantitative comparison between NDTM and other baselines on style guidance generation using Stable Diffusion 1.4. NDTM exhibits better prompt (see CLIP score) and style adherence (see Style Score) over competing baselines.

image. We follow the same setup as MPGD and use a pretrained Stable Diffusion 1.4 text-to-image model. We specify the terminal cost for NDTM in Eq. (22) and compare with MPGD (He et al., 2024) and FreeDOM (Yu et al., 2023) on this task. Figure 3 and Table 4 demonstrate that NDTM exhibits a better balance between prompt and style

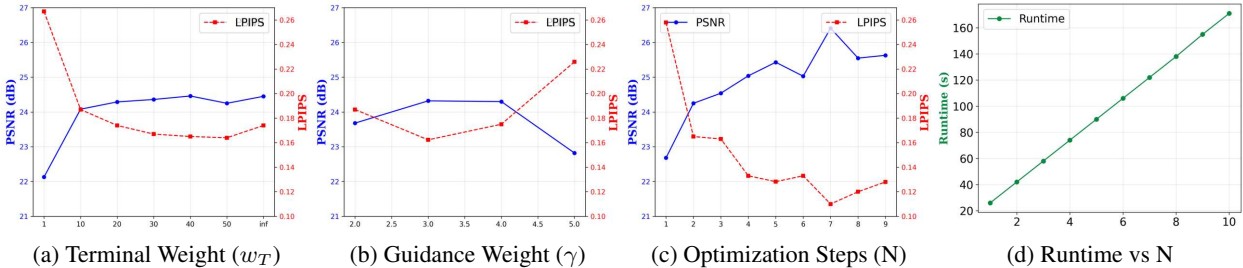

|  |  |  |  |
|---|---|---|---|
| (a) Terminal Weight ($w_T$) | (b) Guidance Weight ($\gamma$) | (c) Optimization Steps (N) | (d) Runtime vs N |

*Figure 4.* **Impact of different design choices** in NDTM on Distortion (PSNR) and Perception (LPIPS) for the non-linear deblur task. (a, b) The extent of guidance can be jointly controlled by varying the terminal loss weight ($w_T$) and the weight ($\gamma$). (c, d) Compute vs quality can be traded off by jointly varying the number of optimization steps (N) and the number of diffusion steps.

adherence over competing baselines.

### 5.3. Ablation Studies

Next, we analyze the impact of different design choices in NDTM on the perception (LPIPS) and distortion (PSNR) quality for the non-linear deblur task on ImageNet.

**Impact of Guidance.** Since the terminal cost weight $w_T$ and the parameter $\gamma$ affect the optimization of the variational control parameters $\mathbf{u}_t$, we analyze their impact on sample quality. From Fig. 4a, we observe that increasing the terminal weight $w_T$ leads to an improvement in both perceptual and distortion quality. However, in the limit of $w_T \to \infty$ (i.e., where the regularization terms in Eq. 17 can be ignored), the perceptual quality degrades, which highlights the importance of the transient cost in our framework. Similarly, increasing $\gamma$ also leads to an improved sample quality. However, a large $\gamma$ can also lead to overshooting.

**Impact of Optimization Steps.** It is common to trade sample quality for the number of sampling steps in diffusion models. Interestingly, NDTM provides a complementary axis to achieve this tradeoff in the form of adjusting the number of optimization steps per diffusion update. We illustrate this in Fig. 4(b), where for a fixed sampling budget of 50 diffusion steps, NDTM can achieve better reconstruction quality by increasing the number of optimization steps (N). However, since the runtime increases linearly as N grows (see Figure 4(d)), a practical choice depends on the available compute. We find that for this task, N=2 provides a favorable tradeoff between sampling time and quality and, therefore, use it for state-of-the-art comparisons on the ImageNet dataset in Table 1.

**Control Visualizations** We visualize the optimal controls $\mathbf{u}_t^*$ in Figure 5. We observe a hierarchical refinement of image features over time. More specifically, the control inference captures global structure at the start of diffusion sampling and gradually refines local details (like edges), thereby encoding high-frequency information at later steps.

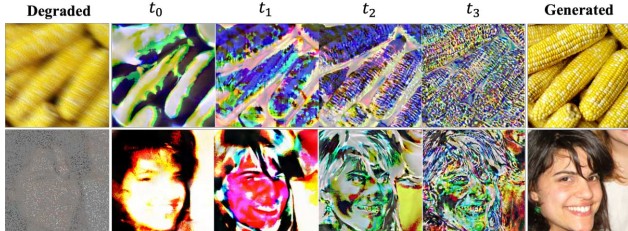

*Figure 5.* **The optimal variational controls hierarchically refine image features over time**. (Top Row) Non-Linear Deblur (Bottom Row) Random Inpainting. (Left to Right) We visualize optimal controls at different times $t_0 > t_1 > t_2 > t_3$ in diffusion sampling, progressively capturing coarse to fine details.

### 6. Conclusion

Our proposed framework offers a principled way to guide a pretrained diffusion model while respecting an external cost through the lens of variational optimal control. Our empirical results suggest that optimizing each diffusion step allows for more flexibility in guidance compared to commonly used approximations of the diffusion posterior.

While our method adapts well to diverse tasks, there remain several interesting directions for future work. First, a more thorough theoretical investigation into the optimization dynamics of the proposed method and integration with existing methods for fast diffusion sampling (Pandey et al., 2024a) could help alleviate the sampling costs of our method. Second, our method is only one instantiation of our framework, leaving ample room for exploration in investigating novel variational parameterizations and refining cost functions which could further enhance the flexibility of our approach.

Lastly, in a broader sense, our work gives another example of the intricate connections between test-time adaptation of diffusion models and Bayesian inference; this view may enable future generative models to perform more complex inference tasks such as hierarchical modeling, and quantifying their uncertainty (Jazbec et al., 2025).

## Acknowledgments

We thank Justus Will for additional discussions and feedback. This project was funded through support from the Chan Zuckerberg Initiative. Additionally, Stephan Mandt acknowledges funding from the National Science Foundation (NSF) through an NSF CAREER Award IIS-2047418, IIS-2007719, the NSF LEAP Center, and the Hasso Plattner Research Center at UCI. Parts of this research were supported by the Intelligence Advanced Research Projects Activity (IARPA) via Department of Interior/ Interior Business Center (DOI/IBC) contract number 140D0423C0075. The U.S. Government is authorized to reproduce and distribute reprints for Governmental purposes notwithstanding any copyright annotation thereon. Disclaimer: The views and conclusions contained herein are those of the authors and should not be interpreted as necessarily representing the official policies or endorsements, either expressed or implied, of IARPA, DOI/IBC, or the U.S. Government.

## Impact Statement

Despite good sample quality in a limited sampling budget, restoration can sometimes lead to artifacts in the generated sample, which can be undesirable in some domains like medical image analysis. Moreover, since our method relies on defining a terminal cost, it can be used for the synthesis of harmful content using an ill-defined cost objective.

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

## A. Proofs

### A.1. Simplification of the NDTM Objective for DDIM

We restate the theoretical result for convenience.

**Proposition A.1.** *For the diffusion posterior parameterization in Song et al. (2021a), the objective in Eq. 16 can be simplified as (see proof in Appendix A.1),*

$$\mathcal{C} \leq \kappa_t^2 \|\mathbf{u}_t\|_2^2 + \tau_t^2 \|\epsilon_\theta(\bar{\mathbf{x}}_t, t) - \epsilon_\theta(\mathbf{x}_t, t)\|_2^2 + w_T \Phi(\hat{\mathbf{x}}_0^t). \tag{23}$$

*where $\bar{\mathbf{x}}_t = \mathbf{x}_t + \gamma \mathbf{u}_t$ is the guided state and the coefficients $\kappa_t = \frac{\gamma\sqrt{\alpha_{t-1}}}{\sqrt{\alpha_t}}$ and $\tau_t = \sqrt{1 - \alpha_{t-1} - \sigma_t^2} - \frac{\sqrt{\alpha_{t-1}(1-\alpha_t)}}{\sqrt{\alpha_t}}$ are time-dependent scalars.*

*Proof.* In the case of DDIM (Song et al., 2021a), the diffusion posterior is parameterized as (Eqn. 12 in Song et al. (2021a)),

$$\boldsymbol{\mu_\theta}(\mathbf{x}_t, t) = \frac{\sqrt{\alpha_{t-1}}}{\sqrt{\alpha_t}}\mathbf{x}_t + \underbrace{\left[\sqrt{1 - \alpha_{t-1} - \sigma_t^2} - \frac{\sqrt{\alpha_{t-1}(1-\alpha_t)}}{\sqrt{\alpha_t}}\right]}_{=\tau_t} \epsilon_\theta(\mathbf{x}_t, t). \tag{24}$$

where the diffusion noising process is parameterized as $p(\mathbf{x}_t|\mathbf{x}_0) = \mathcal{N}(\sqrt{\alpha_t}\mathbf{x}_0, (1 - \alpha_t)\boldsymbol{I}_d)$ and $\epsilon_\theta(\mathbf{x}_t, t)$ is a pretrained denoiser which models $\mathbb{E}[\epsilon|\mathbf{x}_t]$ and intuitively predicts the amount of noise added to $\mathbf{x}_0$ for a given noisy state $\mathbf{x}_t$ at time t. Additionally, for notational convenience, we denote the coefficient of the denoiser in Eq. 24 as $\tau_t$. Following Song et al. (2021a), the coefficient $\sigma$ is further defined as,

$$\sigma_t = \sqrt{\frac{(1 - \alpha_{t-1})}{(1 - \alpha_t)}\left(1 - \frac{\alpha_t}{\alpha_{t-1}}\right)} \tag{25}$$

It follows that,

$$\boldsymbol{\mu_\theta}(\mathbf{x}_t, t) = \frac{\sqrt{\alpha_{t-1}}}{\sqrt{\alpha_t}}\mathbf{x}_t + \tau_t \epsilon_\theta(\mathbf{x}_t, t) \tag{26}$$

$$\boldsymbol{\mu_\theta}(\mathbf{x}_t + \gamma\mathbf{u}_t, t) = \frac{\sqrt{\alpha_{t-1}}}{\sqrt{\alpha_t}}(\mathbf{x}_t + \gamma\mathbf{u}_t) + \tau_t \epsilon_\theta(\mathbf{x}_t + \gamma\mathbf{u}_t, t) \tag{27}$$

$$= \frac{\sqrt{\alpha_{t-1}}}{\sqrt{\alpha_t}}\mathbf{x}_t + \underbrace{\frac{\gamma\sqrt{\alpha_{t-1}}}{\sqrt{\alpha_t}}}_{=\kappa_t}\mathbf{u}_t + \tau_t \epsilon_\theta(\mathbf{x}_t + \gamma\mathbf{u}_t, t) \tag{28}$$

where we denote the coefficient of the control signal $\mathbf{u}_t$ in the above equation as $\kappa_t$ for notational convenience. Consequently, the NDTM cost in Eq. 16 can be simplified for the DDIM posterior parameterization in Eq. 24 as,

$$\mathcal{C} = \left[\|\boldsymbol{\mu_\theta}(\mathbf{x}_t + \gamma\mathbf{u}_t, t) - \boldsymbol{\mu_\theta}(\mathbf{x}_t, t)\|_2^2 + w_T \Phi(\hat{\mathbf{x}}_0^t)\right] \tag{29}$$

$$= \left[\|\kappa_t\mathbf{u}_t + \tau_t(\epsilon_\theta(\mathbf{x}_t + \gamma\mathbf{u}_t, t) - \epsilon_\theta(\mathbf{x}_t, t))\|_2^2 + w_T \Phi(\hat{\mathbf{x}}_0^t)\right] \tag{30}$$

$$\leq^{(i)} \kappa_t^2\|\mathbf{u}_t\|_2^2 + \tau_t^2\|\epsilon_\theta(\mathbf{x}_t + \gamma\mathbf{u}_t, t) - \epsilon_\theta(\mathbf{x}_t, t)\|_2^2 + w_T \Phi(\hat{\mathbf{x}}_0^t) \tag{31}$$

where $(i)$ follows from the triangle inequality. This completes the proof. □

### A.2. Continuous-Time Diffusion Trajectory Matching

Analogous to the discrete case, we represent unguided diffusion dynamics using the following continuous-time reverse diffusion dynamics (Anderson, 1982; Song et al., 2020),

$$d\mathbf{x}_t = \left[f(t)\mathbf{x}_t - g(t)^2 \boldsymbol{s_\theta}(\mathbf{x}_t, t)\right]dt + g(t)d\boldsymbol{w}_t, \tag{32}$$

where $s_{\boldsymbol{\theta}}(\mathbf{x}_t, t)$ is a pretrained score network. Similarly, we parameterize the *guided* continuous dynamics by inserting the control non-linearly into the score function follows,

$$dx_t = \left[ f(t)x_t - g(t)^2 s_{\boldsymbol{\theta}}(x_t, u_t, t) \right] dt + g(t) dw_t. \tag{33}$$

Denote the unguided path measure as $\boldsymbol{u}(\mathbf{x}(T \to 0))$ and the guided path measure as $\mu(\mathbf{x}(T \to 0) | \mathbf{u}(T \to 0))$.

Then, the optimal control problem reads, in analogy to Eq. (8):

$$\mathcal{C}(\mathbf{x}_T, \mathbf{u}(T \to 0)) = w_T \underbrace{\mathbb{E}_\mu[\Phi(\mathbf{x}_0)]}_{\text{Terminal Cost } \mathcal{C}_{\text{te}}} + \underbrace{D_{\text{KL}}(\mu(x(T \to 0)|\mathbf{x}_T, \mathbf{u}(T \to 0)) \,\|\, \nu(x(T \to 0)|\mathbf{x}_T))}_{\text{Transient Cost } \mathcal{C}_{\text{tr}}}. \tag{34}$$

By (Song et al., 2021b, Theorem 1 in Appendix A) (which follows from an application of Girsanov's Theorem), the transient cost reads:

$$\mathcal{C}_{\text{tr}} = D_{\text{KL}}(\mu(x(T \to 0)|\mathbf{x}_T, \mathbf{u}(T \to 0)) \,\|\, \nu(x(T \to 0)|\mathbf{x}_T)) \tag{35}$$

$$= \frac{1}{2} \int g(t)^2 \mathbb{E}_\mu \| s_{\boldsymbol{\theta}}(\mathbf{x}_t, \mathbf{u}_t, t) - s_{\boldsymbol{\theta}}(\mathbf{x}_t, t) \|^2 dt. \tag{36}$$

Taking the approximation that the control signal is optimized greedily, we find Eq. (12).

### A.3. Extension to Flow Matching Models

For continuous flow matching models (Lipman et al., 2023; Albergo & Vanden-Eijnden, 2023; Liu et al., 2023) with a vector field $\boldsymbol{v}_{\boldsymbol{\theta}}(\mathbf{x}_t, t)$,

$$dx_t = \boldsymbol{v}_{\boldsymbol{\theta}}(\mathbf{x}_t, t) dt, \tag{37}$$

we insert the control signal into the dynamics through an additional dependence of the velocity field:

$$\frac{d\mathbf{x}_t}{dt} = \boldsymbol{v}_{\boldsymbol{\theta}}(\mathbf{x}_t, \mathbf{u}_t, t). \tag{38}$$

Since flow matching uses the squared loss, it is natural to regularize deviation from the unguided trajectory in terms of the velocity field:

$$\mathcal{C}_{\text{tr}} = \int \| \boldsymbol{v}_{\boldsymbol{\theta}}(\mathbf{x}_t, \mathbf{u}_t, t) - \boldsymbol{v}_{\boldsymbol{\theta}}(\mathbf{x}_t, t) \|^2 dt \tag{39}$$

## B. Implementation Details

In this section, we include practical implementation details for the results presented in Section 5.

### B.1. Task Details

#### B.1.1. INVERSE PROBLEMS

Here, we describe the task setup in more detail.

**Superresolution (x4)**: We follow the setup from DPS (Chung et al., 2022a), More specifically,

$$\boldsymbol{y} \sim \mathcal{N}(\boldsymbol{y} | \boldsymbol{L}^f \mathbf{x}, \sigma_y^2 \boldsymbol{I}), \tag{40}$$

where $\boldsymbol{S}^f$ represents the bicubic downsampling matrix with downsampling factor $f$. In this work, we fix $f$ to 4 for both datasets.

**Random Inpainting (90%)** We use random inpainting with a dropout probability of 0.9 (or 90%). For this task, the forward model can be specified as,

$$\boldsymbol{y} \sim \mathcal{N}(\boldsymbol{y} | \boldsymbol{M}\boldsymbol{x}, \sigma_y^2 \boldsymbol{I}_d) \tag{41}$$

where $\boldsymbol{M} \in \{0, 1\}^{d \times d}$ is the masking matrix.

**Non-Linear Deblurring** We use the non-linear deblurring setup from DPS. More specifically, we use the forward operator $\mathcal{F}_\phi$ (modeled using a neural network) for the non-linear deblurring operation. Given pairs of blurred and sharp images, $\{\boldsymbol{x}_i, \boldsymbol{y}_i\}$, one can train a forward model estimator as (Tran et al., 2021),

$$\phi^* = \arg\min_\phi \|\boldsymbol{y}_i - \mathcal{F}_\phi(\boldsymbol{x}_i, \mathcal{G}_\phi(\mathbf{x}_i, \mathbf{y}_i))\|_2^2 \tag{42}$$

where $\mathcal{G}$ extracts the kernel information from the training pairs. At inference, the operator $\mathcal{G}$ can instead be replaced by a Gaussian random vector $\boldsymbol{g}$. In this case, the inverse problem reduces to recovering $\boldsymbol{x}_i$ from $\boldsymbol{y}_i$. In this work, we directly adopt the default settings from DPS.

**Blind Image Deblurring (BID)** We directly adopt the setup for blind image deblurring from DMPlug (see Appendix C.4 in Wang et al. (2024) for more details). More specifically, in the BID task, the goal is to recover the kernel $\boldsymbol{k}$ in addition to the original signal $\mathbf{x}_0$ such that,

$$\boldsymbol{y} = \boldsymbol{k} * \boldsymbol{x}_0 + \sigma_y \mathbf{z} \tag{43}$$

In this work, we adapt the default settings from DMPlug. For BID (Gaussian), the kernel size is $64 \times 64$ with the standard deviation set to 3.0. For BID (Motion), the kernel intensity is adjusted to 0.5.

### B.1.2. STYLE GUIDANCE WITH STABLE DIFFUSION

In the context of text-to-image models like Stable Diffusion, the goal of style guidance is to generate a sample that simultaneously adheres well to a given text prompt and style features from a reference image. More specifically, given a reference style image $\boldsymbol{r}$, a pretrained feature extractor (like CLIP) $\mathcal{F}$, a pretrained decoder $\mathcal{D}$ such that $\hat{\mathbf{x}}_0^t = \mathcal{D}(\hat{\mathbf{z}}_0^t)$, we define $\Phi(\hat{\mathbf{x}}_0^t)$ as,

$$\Phi(\hat{\mathbf{x}}_0^t) = \|\mathcal{G}(\mathcal{F}(\boldsymbol{r})) - \mathcal{G}(\mathcal{F}(\hat{\mathbf{x}}_0^t))\|_F^2 \tag{44}$$

where $\mathcal{G}$ denotes the Gram-matrix operation and $\|.\|_F$ denotes the Frobenius norm. In this formulation, the pretrained text-to-image diffusion model works as a generative prior, $p(\mathbf{x}|\mathbf{t})$, where $\mathbf{t}$ is a text prompt embedding, and the goal is to generate samples from the posterior $p(\mathbf{x}|\mathbf{r}, \mathbf{t})$

### B.2. Inverse Problems - Task Specific Hyperparameters

Here, we provide a detailed overview of different hyperparameters for the baselines considered in this work for the inverse problem tasks. We optimize all baselines and our method for the best sample perceptual quality. We use the official code implementation for RED-Diff (Mardani et al., 2024) at `https://github.com/NVlabs/RED-diff`, `https://github.com/mandt-lab/c-pigdm`, and `https://github.com/sun-umn/DMPlug` for running all competing baselines.

### B.2.1. DPS (CHUNG ET AL., 2022A)

We adopt the DPS parameters from Mardani et al. (2024). More specifically, we fix the number of diffusion steps to 1000 using the DDIM sampler. We set $\eta = 0.5$ for all tasks. Following Chung et al. (2022a), we set,

$$\zeta = \frac{\alpha}{\|\boldsymbol{y} - \mathcal{A}(\hat{\mathbf{x}}_0)\|_2^2} \tag{45}$$

Table 5 illustrates different hyperparameters for DPS on all tasks for the FFHQ and ImageNet datasets.

### B.2.2. DDRM (KAWAR ET AL., 2022)

Following Kawar et al. (2022), we fix $\eta = 0.85$, $\eta_b = 1.0$, and use the DDIM sampler with the number of diffusion steps set to 20 across all linear inverse problems.

### B.2.3. C-ΠGDM (PANDEY ET AL., 2024B)

We set the number of diffusion steps to 20 for all tasks. It is also common to contract the reverse diffusion sampling for better sample quality by initializing the noisy state as proposed in Chung et al. (2022b). We denote the start time as $\tau$. We re-tune C-ΠGDM for the best perceptual quality for all linear inverse problems. Table 6 illustrates different hyperparameters for linear inverse problems. We find that C-ΠGDM fails to recover plausible images for the random inpainting task after numerous tuning attempts.

Table 5. DPS hyperparameters for different tasks

|  | FFHQ | ImageNet |
|---|---|---|
| Task | $\alpha$ | $\alpha$ |
| Super-Resolution (4x) | 1.0 | 1.0 |
| Random Inpainting (90%) | 1.0 | 1.0 |
| Non-Linear Deblur | 0.3 | 1.0 |

Table 6. C-ΠGDM hyperparameters used for different tasks. We find that C-ΠGDM fails to recover plausible images for the random inpainting task after numerous tuning attempts.

|  | FFHQ | | | ImageNet | | |
|---|---|---|---|---|---|---|
| Task | $\lambda$ | $w$ | $\tau$ | $\alpha$ | $w$ | $\tau$ |
| Super-Resolution (4x) | -0.4 | 4.0 | 0.4 | -0.4 | 4.0 | 0.4 |
| Random Inpainting (90%) | - | - | - | - | - | - |

Table 7. RED-Diff hyperparameters used for different tasks.

|  | FFHQ | | ImageNet | |
|---|---|---|---|---|
| Task | lr | $\lambda$ | lr | $\lambda$ |
| Super-Resolution (4x) | 0.5 | 1.0 | 0.5 | 0.4 |
| Random Inpainting (90%) | 0.5 | 0.25 | 0.5 | 0.25 |
| Non-Linear Deblur | 0.5 | 0.25 | 0.5 | 0.25 |

### B.2.4. RED-DIFF (MARDANI ET AL., 2024)

We set $\sigma_0 = 0$ with a linear weighting schedule and $lr = 0.5$, $\lambda = 0.25$, and perform 50 diffusion steps using the DDIM sampler for all tasks across the FFHQ and ImageNet datasets. We highlight different hyperparameters in Table 7.

### B.2.5. MPGD (HE ET AL., 2024)

Following the optimal settings in He et al. (2024), we use the setting MPGD w/o proj using the DDIM sampler with 100 diffusion steps and guidance scale set to 5.0 for all tasks.

### B.2.6. NDTM (OURS)

We use the Adam optimizer (Kingma & Ba, 2017) with default hyperparameters, fixing the learning rate to 0.01 for updating the control $\mathbf{u}_t$ across all tasks and fixing the kernel learning rate in the BID task to 0.01. We refer to the loss weighting scheme in Eq. 16 as "DDIM weighting". Moreover, we use linear decay for the learning rate. We perform 50 diffusion steps using the DDIM sampler across all datasets and tasks. We tune the guidance weight $\gamma$, the number of optimization steps N, loss weighting ($w_T$, $w_s$, $w_c$), DDIM $\eta$ and the truncation time $\tau$ (Chung et al., 2022b) for best performance across different tasks. All these hyperparameters are listed in Table 9.

### B.2.7. RB-MODULATION (ROUT ET AL., 2025)

Since RB-Modulation is a special case of NDTM with $\gamma = 1.0$ and $w_s = w_c = 0$, we re-run NDTM for different tasks with these settings, keeping all other hyperparameters fixed to report results for RB-Modulation.

### B.3. Style Guidance Hyperparameters

### B.3.1. MPGD (HE ET AL., 2024)

Following He et al. (2024), we use the DDIM sampler with 100 steps and $\eta = 1.0$ without the time reversal (Lugmayr et al., 2022) trick for fair comparisons. We set $\rho = 17.5$ and the classifier-free guidance scale to 7.5

### B.3.2. FREEDOM (YU ET AL., 2023)

Following Yu et al. (2023), we use the DDIM sampler with 100 steps and $\eta = 1.0$ without the time reversal trick. We use the classifier-free guidance scale of 5.0

*Table 8.* BID hyperparameters for NDTM.

| | | FFHQ | | | | | |
|---|---|---|---|---|---|---|---|
| Task | N | $\gamma$ | $\eta$ | $\tau$ | $w_T$ | $w_s$ | $w_c$ |
| BID (Gaussian) | 15 | 1.0 | 0.7 | 1000 | 50 | ddim | ddim |
| BID (Motion) | 15 | 1.0 | 0.7 | 1000 | 50 | ddim | ddim |

*Table 9.* NDTM hyperparameters for different tasks.

| | | | FFHQ | | | | | | | ImageNet | | | | | |
|---|---|---|---|---|---|---|---|---|---|---|---|---|---|---|---|
| Task | N | $\gamma$ | $\eta$ | $\tau$ | $w_T$ | $w_s$ | $w_c$ | N | $\gamma$ | $\eta$ | $\tau$ | $w_T$ | $w_s$ | $w_c$ |
| Super-Resolution (4x) | 5 | 1.0 | 0.7 | 400 | 50 | ddim | ddim | 2 | 2.0 | 0.1 | 600 | 50 | ddim | ddim |
| Random Inpainting (90%) | 2 | 4.0 | 0.2 | 500 | 1 | 0 | 0 | 2 | 4.0 | 0.0 | 600 | 50 | ddim | ddim |
| Non-Linear Deblur | 5 | 5.0 | 0.1 | 400 | 1 | 0 | 0 | 2 | 4.0 | 0.1 | 600 | 50 | ddim | ddim |

### B.3.3. NDTM

For NDTM, we use the DDIM sampler with 50 steps and $\eta = 1.0$. We set the control learning rate to 0.002 with the Adam optimizer. The loss weightings are set to $w_T = 1.0, w_c = 0, w_s = 0$ with $\gamma = 4.0, N = 2$ and a classifier-free guidance scale of 5.0

## C. Additional Experimental Results

### C.1. Evaluation on Distortion Metrics

In this work, we primarily optimize all competing methods for perceptual quality. However, for completeness, we compare the performance of our proposed method with other baselines on recovery metrics like PSNR and SSIM. Tables 12 and 13 compare our proposed method, NDTM, with competing baselines for linear and non-linear inverse problems. We find that NDTM performs on par with other methods for the super-resolution task. However, for random inpainting and non-linear deblur, NDTM outperforms competing methods in terms of distortion metrics like PSNR. Since NDTM also outperforms existing baselines in terms of perceptual quality (see Table 2), our method provides a better distortion-perception tradeoff.

### C.2. Runtime

Below, we compare different methods in terms of the wall-clock time required for running on a single image for the superresolution task. From Table 10, we observe that while our method requires an inner optimization loop, it is still faster than common baselines like DPS and DMPlug (see Table 11).

*Table 10.* Runtime comparisons for different baselines vs NDTM for super-resolution task on both datasets. The runtime numbers are in wall-clock time (seconds) and tested on a single RTX A6000 GPU.

| | FFHQ ($256 \times 256$) | | | | | Imagenet ($256 \times 256$) | | | | |
|---|---|---|---|---|---|---|---|---|---|---|
| | DPS | RED-diff | C-ΠGDM | DDRM | NDTM (Ours) | DPS | RED-diff | C-ΠGDM | DDRM | NDTM (Ours) |
| Runtime (secs / Img) | 199.1 | 5.8 | 3.68 | 1.3 | 13.6 | 399.3 | 7.1 | 16.4 | 2.4 | 38.3 |

*Table 11.* Runtime comparisons for DMPlug baseline vs NDTM for blind image deblurring (BID) task on FFHQ dataset. The runtime numbers are in wall-clock time (minutes) per image and tested on a single RTX A6000 GPU.

| Method | Gaussian blur | Motion blur |
|---|---|---|
| | Time↓ | Time↓ |
| DMPlug | 51.24 | 51.13 |
| NDTM$^{\dagger}$ (Ours) | **7.17** | **7.17** |
| NDTM$^{\zeta}$ (Ours) | 18.07 | 18.13 |

*Table 12.* Comparison between NDTM and existing methods for Linear IPs on distortion metrics like PSNR and SSIM. Missing entries indicate that the method was unstable for that specific task. **Bold**: best.

| | Super-Resolution (4x) | | | | Random Inpainting (90%) | | | |
|---|---|---|---|---|---|---|---|---|
| | FFHQ (256 × 256) | | Imagenet (256 × 256) | | FFHQ (256 × 256) | | Imagenet (256 × 256) | |
| Method | PSNR↑ | SSIM↑ | PSNR↑ | SSIM↑ | PSNR↑ | SSIM↑ | PSNR↑ | SSIM↑ |
| DPS | 29.06 | 0.832 | 23.61 | 0.676 | 27.76 | 0.832 | 20.96 | 0.657 |
| DDRM | **30.12** | **0.864** | **24.15** | **0.701** | 17.34 | 0.371 | 15.91 | 0.257 |
| RED-diff | 27.67 | 0.720 | 24.06 | 0.685 | 20.84 | 0.581 | 18.63 | 0.466 |
| C-ΠGDM | 27.93 | 0.773 | 23.20 | 0.631 | - | - | - | - |
| MPGD | 26.07 | 0.715 | 21.83 | 0.587 | 11.34 | 0.076 | 10.26 | 0.025 |
| RB-Modulation | 29.12 | 0.831 | 23.41 | 0.674 | 26.90 | 0.810 | 21.31 | 0.632 |
| NDTM (ours) | 29.06 | 0.833 | 23.12 | 0.674 | **28.03** | **0.834** | **21.34** | **0.665** |

*Table 13.* NDTM outperforms existing methods for Non-linear deblur on distortion metrics like PSNR and SSIM. **Bold**: best.

| | FFHQ (256 × 256) | | ImageNet (256 × 256) | |
|---|---|---|---|---|
| Method | PSNR↑ | SSIM↑ | PSNR↑ | SSIM↑ |
| DPS | 8.12 | 0.262 | 6.67 | 0.156 |
| RED-diff | 24.88 | 0.717 | 21.88 | 0.623 |
| MPGD | 18.24 | 0.406 | 17.02 | 0.261 |
| RB-Modulation | 29.39 | 0.846 | 22.11 | 0.612 |
| NDTM (ours) | **30.64** | **0.874** | **24.41** | **0.732** |

| Original | Degraded | NDTM (Ours) | RED-Diff | RB-Modulation |
|----------|----------|-------------|----------|---------------|

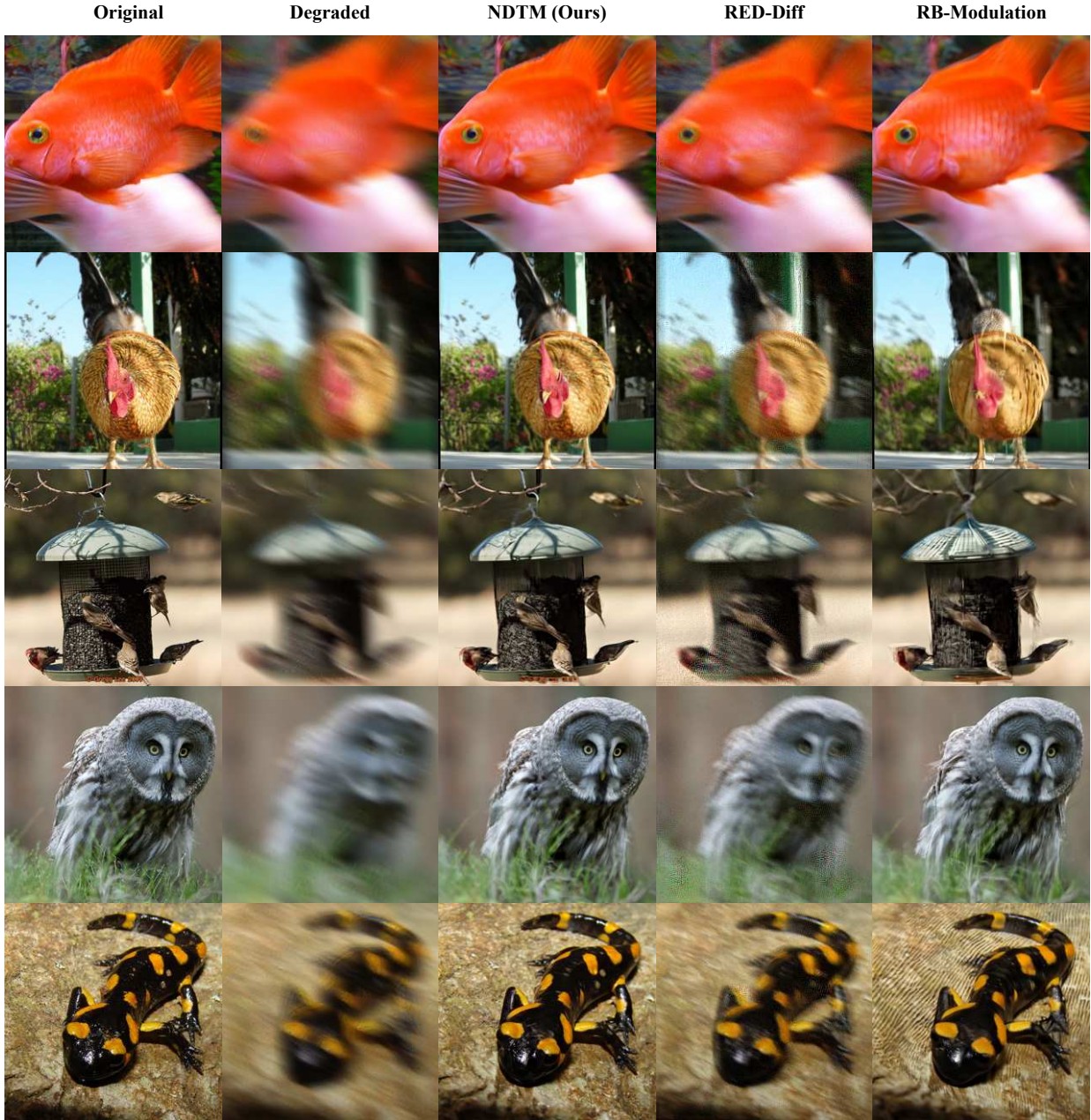

*Figure 6.* Qualitative comparison between NDTM and top competing baselines (see Table 1) on the Non-Linear Deblurring task for ImageNet. NDTM better recovers the structure of the image compared to other baselines. Best viewed when zoomed in.

| Original | Degradation | NDTM (Ours) | MPGD | Red-Diff | RB-Modulation |
|---|---|---|---|---|---|

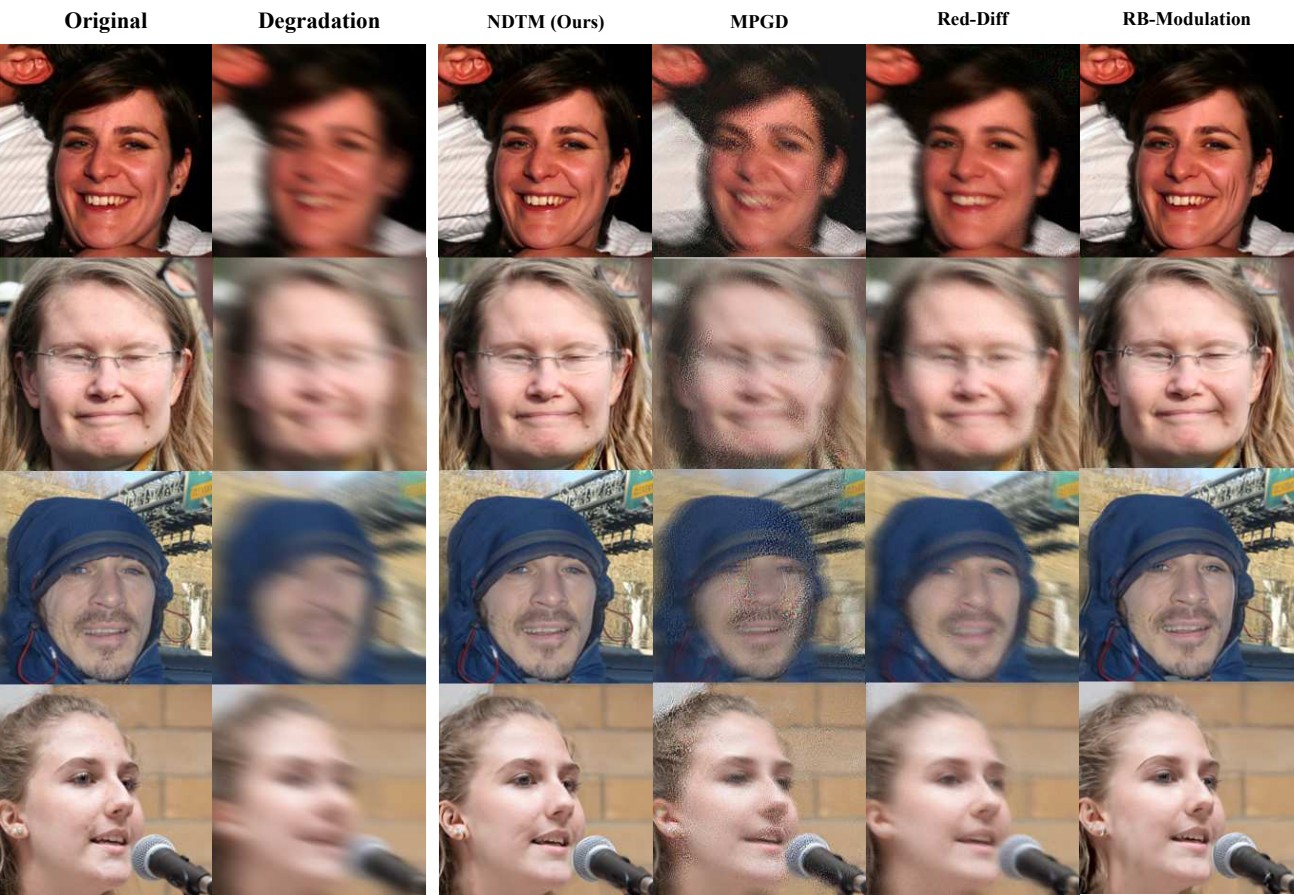

*Figure 7.* Qualitative comparison between NDTM and top competing baselines (see Table 1) on the Non-Linear Deblurring task for FFHQ. NDTM better recovers the structure of the image compared to other baselines. Best viewed when zoomed in.

| Original | Degraded | NDTM (Ours)
(M=15, T=400) | NDTM (Ours)
(M=15, T=200) | DMPlug |
| --- | --- | --- | --- | --- |

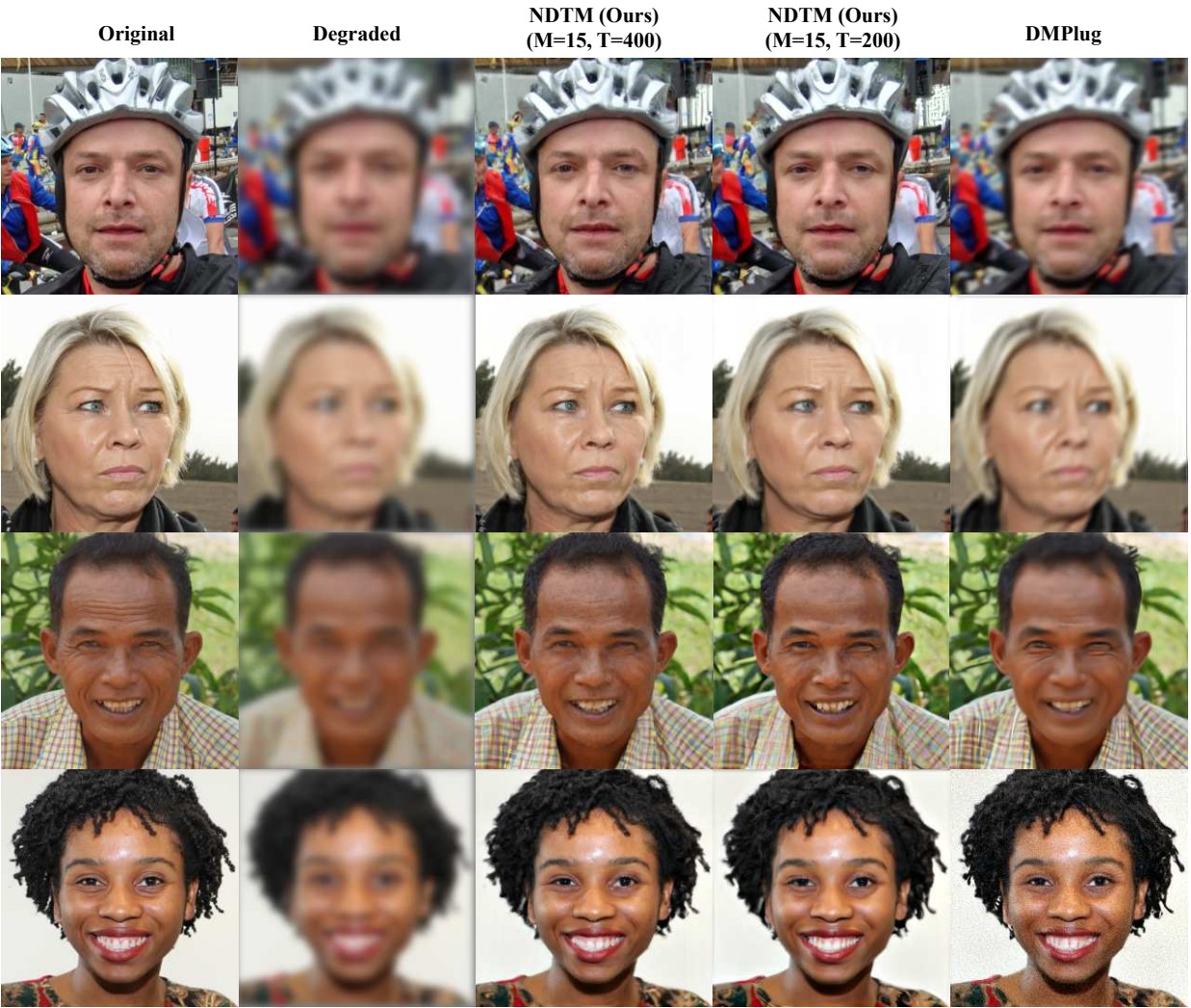

*Figure 8.* Qualitative comparison between NDTM and competing baseline (DMPlug) on the blind image deblurring task (see Table 3). NDTM better recovers the details and structure of the image compared to the baseline. We find DMPlug introduces noisy artifacts and blurry images in some samples. Best viewed when zoomed in.

| Original | Degraded | NDTM (Ours) | DPS | RED-Diff | RB-Modulation |
|---|---|---|---|---|---|

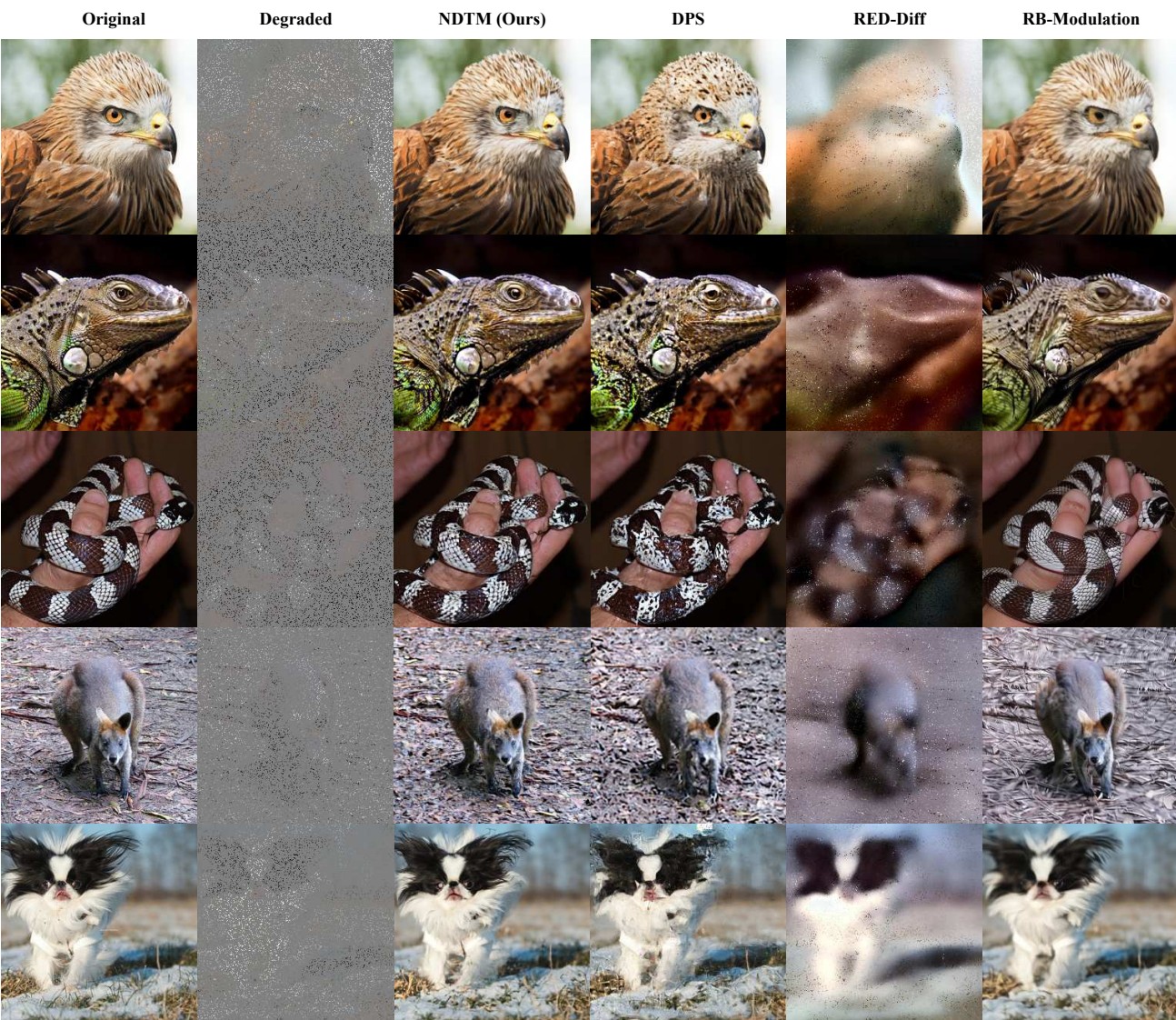

*Figure 9.* Qualitative comparison between NDTM and top competing baselines (See Table 2) on the Random Inpainting (90%) Task for ImageNet. NDTM performs better or/par compared to other baselines. Best viewed when zoomed in.

| **Original** | **Degradation** | **NDTM (Ours)** | **Red-Diff** | **RB-Modulation** | **DPS** |

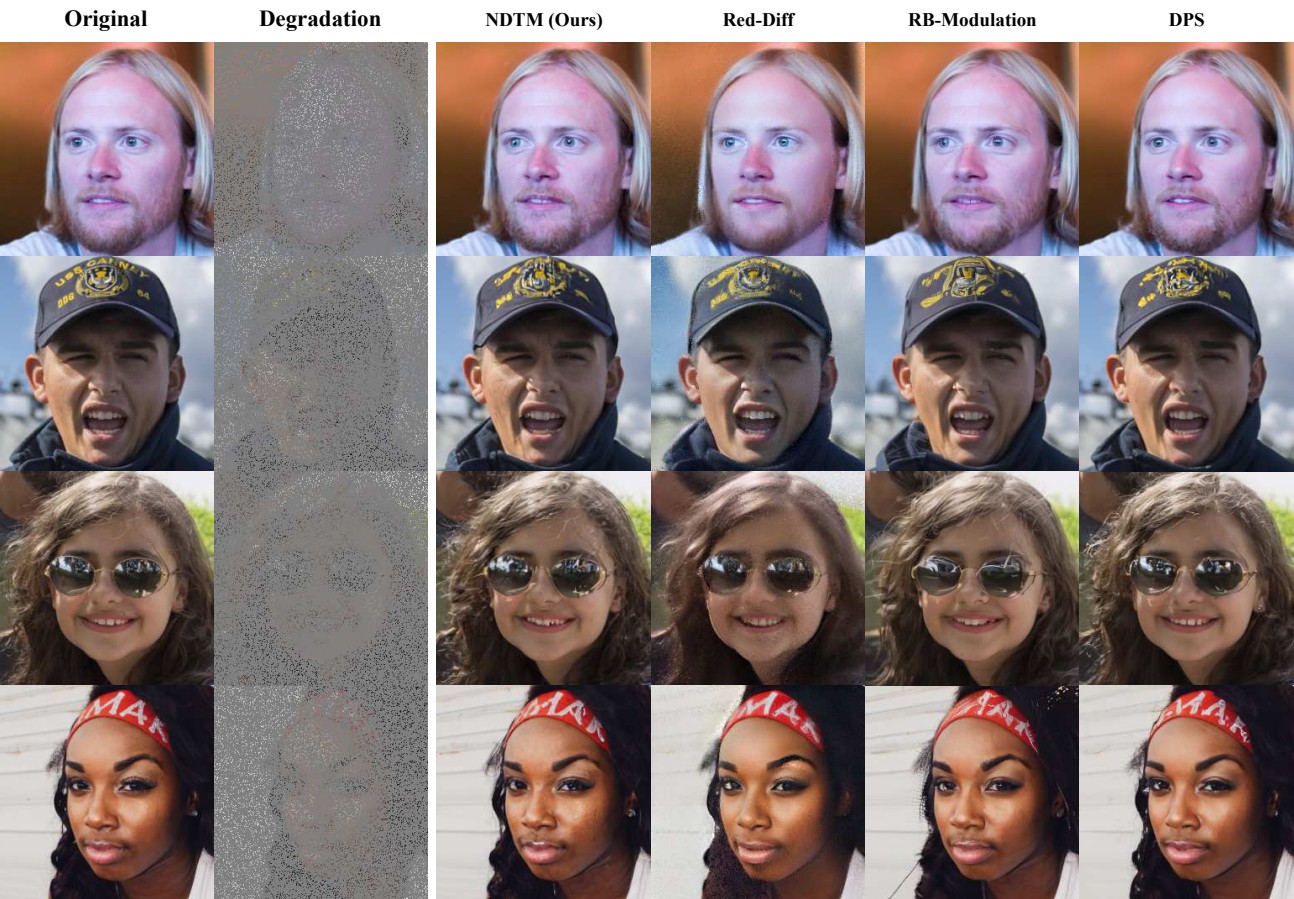

*Figure 10.* Qualitative comparison between NDTM and top competing baselines (See Table 2) on the Random Inpainting (90%) Task for FFHQ. NDTM performs better or/par compared to other baselines. Best viewed when zoomed in.

| Original | Degraded | NDTM (Ours) | DPS | C-ΠGDM | RB-Modulation |
|----------|----------|-------------|-----|--------|---------------|

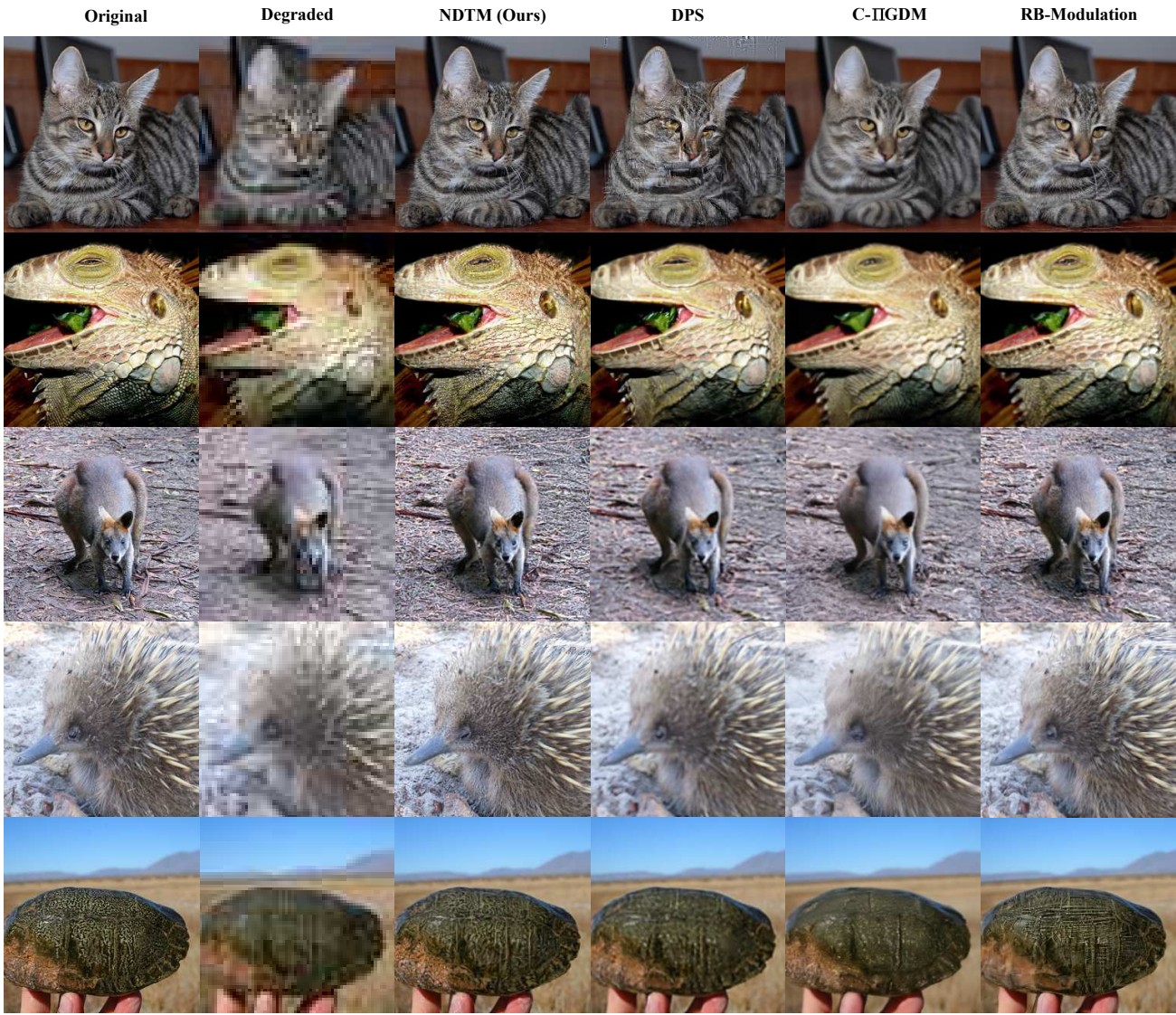

*Figure 11.* Qualitative comparison between NDTM and top competing baselines (See Table 2) on 4x super-resolution task for ImageNet. NDTM performs better or/par compared to other baselines. Best viewed when zoomed in.

| Original | Degradation | NDTM (Ours) | RB-Modulation | MPGD | C-ΠGDM | DPS |
|----------|-------------|-------------|---------------|------|--------|-----|

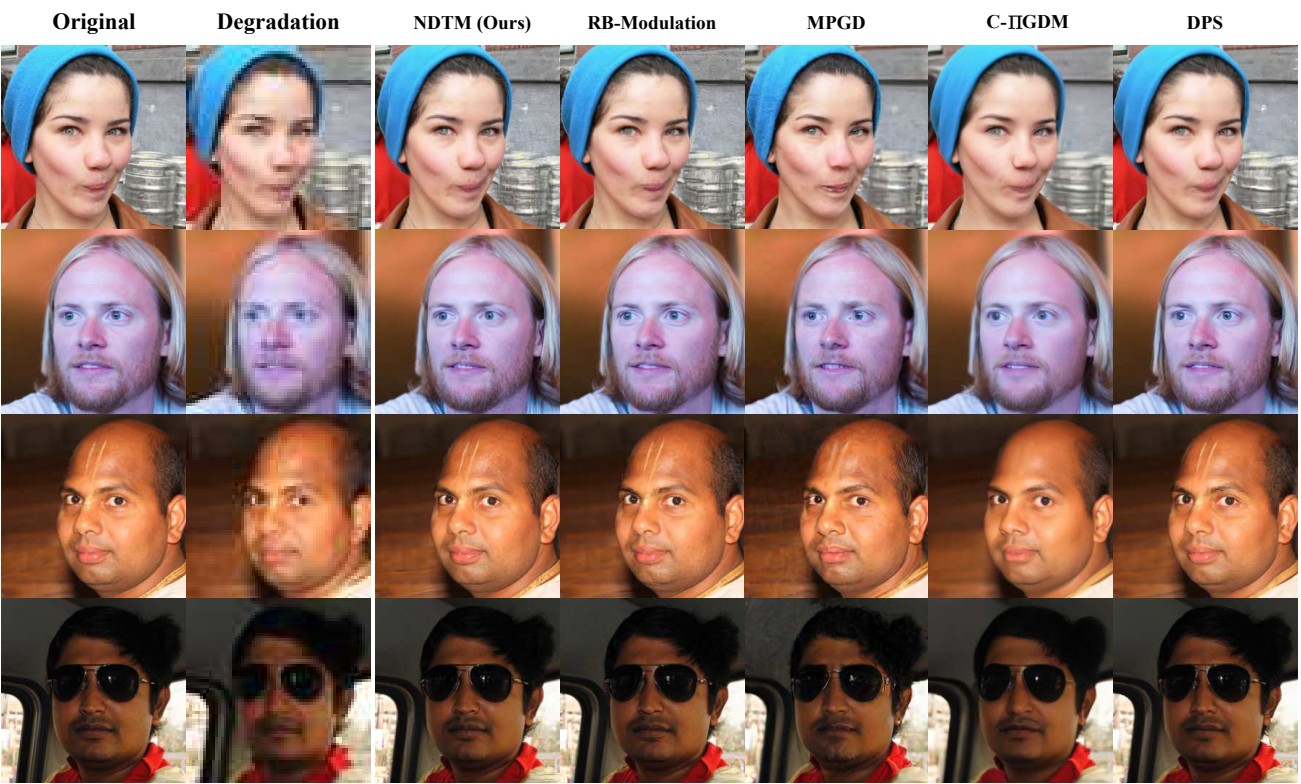

*Figure 12.* Qualitative comparison between NDTM and top competing baselines (See Table 2) on 4x super-resolution task for FFHQ. NDTM performs better or/par compared to other baselines. Best viewed when zoomed in.

