# OpenReview forum: "Variational Control for Guidance in Diffusion Models"
_ICML.cc/2025/Conference — ICML 2025 poster_

### Official Review · Reviewer_amRw · 2025-02-25

**Overall Recommendation:** 4

**Summary:**

This paper introduces DTM - Diffusion Trajectory Matchning - a novel and general guidance approach for generic diffusion models. The idea is to add a guidance vector at each time point, u_t, such that it serves the given measurements while also maikg sure that the original diffusion trajectory is not deviated much. The vector u_t is found by minimizing a cost function that takesthe above two forces into account, and then it is added as the guidance for the current stage. When simplified, this approach amounts to a guidance that is similar to DPS (but avoiding the derivative through the denoiser) , but with a regularization that forces a proximity between the means of the conditional probability means (\mu(x_t) versus \mu(xt+\gamma*u_t)). This last result leads to a non-linear computaiton of u_t, termed NDTM (Non-Linear Diffusion Trajectory Matching). A further simplified version of this is derived for DDIM, penalyzing for the length of u_t, and the difference between the two denoiser outputs. Various experiments demonstrate the superiority of this approach over the alternative methods in handling inverse problems, linear and non-linear ones.

**Claims And Evidence:**

Excellent paper

**Essential References Not Discussed:**

None.

**Experimental Designs Or Analyses:**

Very well designed experiments

**Methods And Evaluation Criteria:**

Perfect.

**Other Comments Or Suggestions:**

Looking at Eq. (17), there are 3 forces that are involved in the creation of ut:
1. Forcing the norm of this vector to be small (||ut||_2)
2. Forcing the trajectory not to deviate much by forcing the two denoiising results to be close by, and
3. Forcing the measurements in some sort of projection step on the denoised image.

Of these three, #2 complicates things and makes the method non-linear and more complicated for optimization. Therefore, an ablation that shows the effect of excluding #2 is necessary. Perhaps #1 would be sufficient to regularize the guidance. Furthyeremore, for linear inverse problems, ut will have a closed form solution of #2 is omitted.

**Other Strengths And Weaknesses:**

Nothing to add. This paper is pleasant to read.

**Questions For Authors:**

None.

**Relation To Broader Scientific Literature:**

Explains well the surrounding papers, including stochastric control methods that inspired this work.

**Theoretical Claims:**

All are correct.

---

> ### Author Rebuttal · Authors · 2025-04-01
>
> We thank the reviewer for their encouraging feedback about experiment design and readability. We address specific questions below:
>
> > Of these three, #2 complicates things and makes the method non-linear and more complicated for optimization. Therefore, an ablation that shows the effect of excluding #2 is necessary. Perhaps #1 would be sufficient to regularize the guidance. Furthyeremore, for linear inverse problems, ut will have a closed form solution of #2 is omitted.
>
> We would like to thank the reviewer for their insight. We agree that such an ablation could be useful to study the impact of different terms in the loss function in Eq. 17. Empirically, we find the choice of weighting to be largely task-specific (see Tables 7 and 8).

---

> > ### Comment · Reviewer_amRw · 2025-04-03
> >
> > The grade remains as is after the response.

---

### Official Review · Reviewer_UY5V · 2025-03-14

**Overall Recommendation:** 2

**Summary:**

The paper introduces a framework called Diffusion Trajectory Matching (DTM) for guiding diffusion models without requiring retraining. This approach is rooted in variational inference and optimal control, allowing for guidance by optimizing control signals based on terminal costs. The proposed method, Non-linear Diffusion Trajectory Matching, integrates with existing samplers like DDIM and targets performances on linear and non-linear inverse problems by optimizing diffusion trajectories toward a desired outcome, showing improvements in metrics like FID over some baselines.

**Claims And Evidence:**

The claims made in the submission are generally right.

**Essential References Not Discussed:**

The paper lacks a comprehensive literature review. There are many guidance methods in diffusion which the authors did not discuss or compare, such as [1-8].

[1] Unlocking guidance for discrete state-space diffusion and flow models.

[2] Monte carlo guided diffusion for bayesian linear inverse problems.

[3] Amortizing intractable inference in diffusion models for vision, language, and control

[4] Derivative-Free Guidance in Continuous and Discrete Diffusion Models with Soft Value-Based Decoding

[5] Steering Masked Discrete Diffusion Models via Discrete Denoising Posterior Prediction

[6] Alignment without Over-optimization: Training-Free Solution for Diffusion Models

[7] Diffusion Model Alignment Using Direct Preference Optimization

[8] DyMO: Training-Free Diffusion Model Alignment with Dynamic Multi-Objective Scheduling

**Experimental Designs Or Analyses:**

There lack experimental comparisons with more recent methods. Please refer to Essential References Not Discussed.

**Methods And Evaluation Criteria:**

The evaluation tasks make sense but the comparisons are limited.

**Other Comments Or Suggestions:**

Please refer to above.

**Other Strengths And Weaknesses:**

Strengths:
1. The integration of variational control into the guidance of diffusion models is well-founded.
2. Demonstrated improvements on tasks such as non-linear and blind inverse problems show broader practical applications.

Weaknesses:
1. The paper does not provide code for verification.
2. For the comparison with DPS, the paper seems to fix the hyperparameters. Given that the method itself's performance heavily depends on the correct setting of hyperparameters such as guidance weight and terminal cost weight, hyperparameters of DPS should also be selected from a space.
3. The paper lacks a comprehensive literature review. There are many training-free guidance methods in diffusion which the authors did not discuss or compare. Please refer to Essential References Not Discussed.
4. While the paper discusses non-linear control benefits, comparisons primarily focus on linear control setups, possibly overlooking the full spectrum of non-linear dynamics.
5. The method assumes differentiable objectives, which may not be applicable or ideal for all types of generative tasks.
6. The paper lacks Impact Statements, which is required.

**Questions For Authors:**

Please refer to above.

**Relation To Broader Scientific Literature:**

The paper discussed the relation of its idea with Classifier Guidance in Diffusion Models and Optimal Control. The key contributions of the paper relate well to the broader scientific literature by extending the capabilities of diffusion models in inverse problem-solving.

**Theoretical Claims:**

The paper doesn't include theorems. The Proofs section in the submission is essentially formulations instead of rigorous math proofs.

---

> ### Author Rebuttal · Authors · 2025-04-01
>
> We thank the reviewer for their feedback. Please find our responses below.
>
> > The evaluation tasks make sense but the comparisons [...] lack [...] more recent methods. . Please refer to Essential References Not Discussed.[...] authors did not discuss or compare, such as [1-8].
>
> This was also a common point from Reviewers DxJn, 113Q so we address it jointly here.
>
> **Re. Generalization to other tasks**. Our framework indeed generalizes to non-linear tasks such as those pointed out by the reviewers. We demonstrate this by simply adapting the terminal cost in our method for Style and text-conditioned guidance. In more detail:
>
> **Style Guidance**: Following the experimental setup in MPGD [He et al.], we evaluate our method on Style Guidance using Stable Diffusion (see [Figure 1](https://imgur.com/a/wEDqoaR)). We further quantitatively compare our method on 1k (reference images, prompts) with FreeDOM [Yu et al.] and MPGD in [Table 1](https://imgur.com/a/jTn0hQ2). Our quantitative and qualitative results suggest that NDTM outperforms competing baselines on prompt (CLIP score) and style adherence (Style Score) while exhibiting better perceptual quality.
>
> **Text Guidance with CLIP**: Following the experimental setup in MPGD, we present qualitative results on text-conditional generation on the CelebA (see [Figure 2](https://imgur.com/a/5jlnDLo). NDTM can be used to generate samples that adhere well to simple and complex prompts.
>
> Therefore, we emphasize that our method can be applied to more general guidance scenarios. We will include additional qualitative results for these tasks with their experimental settings in a revised version of our paper. Note that we selected the best hyperparameters for the selected baselines, just like for all other experiments.
>
> **Re. Limited Empirical Comparisons**. Based on the suggestion from Reviewer 113Q, we compare with two additional baselines, MPGD and RB-Modulation [Rout et al.], on the non-linear deblur and 4x superresolution tasks for the FFHQ and ImageNet datasets. NDTM outperforms these additional baselines on these tasks. See [Table 2](https://imgur.com/a/Y8Mlvw0).
>
> We also thank the reviewer for the interesting pointers, which we will include in the related work section for broader context. However, they primarily focus on discrete diffusion models or finetuning with human preference data—directions that differ from the scope and goals of our paper. In detail:
>
> **Re. Discrete Diffusion Models.** Our work focuses on continuous state space diffusion models (see Section 2) and assumes a differentiable terminal cost (see line 120, Right Column in the main text). We think that optimal control provides an interesting approach to guiding discrete diffusion models and leave this open to future work.
>
> **Re. Finetuning with human preference data.** In this work, we only work with training-free methods for conditional generation tasks and not with methods that finetune diffusion models with additional human preference data. Therefore, comparisons with baselines like Diffusion-DPO are outside the scope of this work.
>
> > The paper does not provide code for verification
>
> We will make our code publicly available upon acceptance. Please refer to the pseudocode in Algorithm 1 in the main text and the corresponding hyperparameters in Tables 4,5 and 6 in the Appendix.
>
> > For the comparison with DPS [...]  should also be selected from a space.
>
> We tuned all baselines (including DPS) for the best performance for all tasks and datasets (see Appendix B.2).
>
> > While the paper discusses non-linear control [...] of non-linear dynamics.
>
>  We respectfully disagree, as there seems to be a misunderstanding: We both consider non-linear inverse problems and allow for non-linear control. In detail, while we do update the state $x_t$ additively with the control $u_t$, the updated state ($x_t + \gamma u_t$) is directly injected in the diffusion denoiser (modeled as a non-linear neural network). Therefore, the guidance process indeed depends non-linearly on the control. We acknowledge that to update the state $x_t$ with the control $u_t$, there could be multiple ways to achieve this. However, we don’t explore the full space of such possible combinations due to space and time constraints which we also highlight in line 175, Right column.
>
> > The method assumes differentiable objectives [...] generative tasks.
>
> We clearly state our assumptions of a differentiable terminal cost in Line 120, Right Column in the main text. Therefore, while the argument that our method may not be ideal for all types of generative tasks is correct, we never claim this in our work. We agree that extending our method to discrete diffusion models and discrete terminal costs can be an interesting direction for further work and will clarify this point in more detail in Section 6 of our revised paper.
>
> > The paper lacks Impact Statements, which is required.
>
> We have provided the impact statement at line 435 (Right Side) in the main text.

---

### Official Review · Reviewer_113Q · 2025-03-14

**Overall Recommendation:** 3

**Summary:**

In this paper, the authors formulate the diffusion posterior guidance problem as a variational control problem and propose a novel training-free framework for this problem. They introduce a new algorithm for diffusion guidance and their framework also unifies many existing training-free diffusion guidance algorithms. They have conducted experiments on FFHQ-256 and ImageNet-256 datasets to show superior performance in comparison to several popular baselines.

**Claims And Evidence:**

I do find the statement that the authors make about the "broad generalizability" of their method to be overclaiming.

In particular, the authors claim in the introduction that prior works which are based on optimal control only "focus on a restricted class of control problems", and their method "adapts well to diverse downstream tasks". However, in their experiments, the authors only show evidence on simple inverse problems like deblurring and inpainting. The only non-linear task that the authors explore is non-linear deblurring, and none of the more complicated non-linear task such as style guidance generation, face identity guidance generation or text conditioned generation (all of which are widely studied [1,2,3,4]).

Moreover, the authors claim to be able to solve "(blind) non-linear inverse problems". However, the only blind inverse problem that they attempt to solve is linear deblurring. (They assume that there exist a blurring kernel and they would apply linear convolution with this kernel).  As a result, I find the claim about the diversity and the generalizability of the method to be inaccurate and not well justified by the evidence provided.


[1] Bansal, et al. Universal Guidance for Diffusion Models. ICLR 2024.

[2] Yu et al. FreeDoM: Training-Free Energy-Guided Conditional Diffusion Model. ICCV 2023.

[3] He et al. Manifold Preserving Guided Diffusion. ICLR 2024.

[4] Rout et al. RB-Modulation: Training-Free Personalization of Diffusion Models using Stochastic Optimal Control. ICLR 2025.

[5] Ye et al. TFG: Unified Training-Free Guidance for Diffusion Models. NeurIPS 2024.

**Essential References Not Discussed:**

The most prominent concern I have about this paper is the lack of comparison with relevant literature. As I have mentioned earlier in this review, the authors fail to mention and/or compare a large portion of related works in training-free diffusion guidance, many of which achieves better performance than the baselines selected by the authors. In particular, here is a list of papers that the authors should consider comparing to:

[1] Bansal, et al. Universal Guidance for Diffusion Models. ICLR 2024.

[2] Yu et al. FreeDoM: Training-Free Energy-Guided Conditional Diffusion Model. ICCV 2023.

[3] He et al. Manifold Preserving Guided Diffusion. ICLR 2024.

[4] Rout et al. RB-Modulation: Training-Free Personalization of Diffusion Models using Stochastic Optimal Control. ICLR 2025. (The authors have mentioned this work in their paper but failed to compare with this method)

[5] Ye et al. TFG: Unified Training-Free Guidance for Diffusion Models. NeurIPS 2024.

[6] Chung et al. Improving diffusion models for inverse problems using manifold constraints. NeurIPS 2022.

**Experimental Designs Or Analyses:**

My concerns about the experimental design are elaborated in section "Methods And Evaluation Criteria" and "Essential References Not Discussed".

**Methods And Evaluation Criteria:**

The authors have provided sufficient justification and proposed mostly valid evaluation criteria for the problem at hand. However, since the evaluations are conducted with only 1000 samples, FID is not an appropriate metric for these experiments (FID is highly biased when using a small number of samples [1]). Given that the authors also have included the KID results, which is the more appropriate metric to be used, I am only mentioning this because the authors have emphasized their performance gain based on FID results in their abstract.

[1] Binkowski et al. Demystifying MMD GANs. ICLR 2018.

**Other Comments Or Suggestions:**

N/A

**Other Strengths And Weaknesses:**

I have elaborated the strengths and weaknesses of this paper in the previous sections.

**Questions For Authors:**

The authors choose $T=200,N=15$ as the hyperparameters in their experiments. So in total there will be $200\times(15+1)=3200$ diffusion NFE required for each sample. However, their method is significantly faster than DPS, which only requires $1000$ NFE. Can the author explain why they can achieve this speedup?

**Relation To Broader Scientific Literature:**

One of the biggest strength of this paper is that the authors do provide a unified framework that generalizes diffusion posterior sampling method to an optimal control inspired formulation. Solving inverse problems with training-free diffusion guidance is also a vibrant and important research field as it is closely related to imaging processing, broader signal processing and many medical applications.

**Theoretical Claims:**

I have two major concerns regarding the theoretical aspect of this paper.

1. The authors choose addition as the aggregation function to incorporate control for calculating the posterior mean. However, this design choice is not justified at all. As a matter of fact, without any constraint on $u_t$, the diffusion model may not even be well defined on $x_t + \gamma u_t$ because all noisy samples are concentrated on certain intermediate manifolds during the diffusion process [1]. It would be great if the authors can provide further justification and clarification on the assumptions when choosing the aggregation function of the control.

2. The authors mentioned that the Tweedie’s estimate in DPS causes high computational requirement and high sensitivity to the hyperparameters. However, in the proposed approach, the authors also use the Tweedie’s estimate for $\hat{x}_0$. I wonder why the authors consider their method to be exempt from the same limitation.

[1] Chung et al. Improving diffusion models for inverse problems using manifold constraints. NeurIPS 2022.

---

> ### Author Rebuttal · Authors · 2025-04-01
>
> We thank the reviewer for their detailed and helpful feedback. Please see our response below.
>
> > I do find the statement [...] the authors claim in the introduction [...] that prior works which are based on optimal control [...] (all of which are widely studied [1,2,3,4])
>
> Our framework indeed generalizes to non-linear tasks like Style and text guidance. Due to space constraints, please see our response to Reviewer UY5V (first point).
>
> > Moreover, the authors claim to be able to [...] However, the only blind inverse [...] to be inaccurate and not well justified by the evidence provided.
>
> For the blind deblurring task considered in this work, the forward model y = k * x + n is non-linear since both the kernel k and the underlying signal x are unknown here, making the inverse problem non-linear and challenging. Therefore, we disagree that it is equivalent to solving linear deblurring. Moreover, as highlighted in our common response, we already show that our method can generalize to tasks like Style Guidance, providing further evidence in favor of its generalizability.
>
> > The authors have [...] proposed mostly valid evaluation criteria for the problem at hand [...] performance gain based on FID results in their abstract
>
> We agree that 1000 samples may produce a biased FID. However, we evaluate all baselines similarly, so we expect similar improvements in FID for more samples. In addition, the reported FID strongly correlates with other more stable metrics, such as LPIPS and KID. Lastly, our choice of metrics and evaluation protocol is primarily based on prior work in the literature like DPS [Chung et al.] and RedDiff [Mardani et al.] and is therefore justified, as also noted by the reviewer.
>
> > The authors choose addition as the aggregation function [...] further justification and clarification on the assumptions when choosing the aggregation function of the control.
>
> We do regularize u_t via the guidance scale $\gamma$, compare Eq. 17 in the paper. More specifically, the first term in Eq. 17 regularizes the magnitude of u_t, while the second term encourages a u_t which does not change the magnitude of noise in the guided signal x_t. This brings the guided trajectory close to the unguided trajectory and close to the usual input to the learnt score. Regarding the image manifold, since u_t is only added inside a score function, the terminal cost cannot make the trajectory leave the span of the score function. More importantly, superior empirical results across a range of linear and non-linear inverse problems justify our choice of aggregation function. Lastly, we did not explicitly explore specifying terminal costs that are incompatible with the diffusion prior. We think that this is an interesting future direction and will add this to the Limitations in Section 6 of the revised paper. We hope this clarifies any concerns about our choice of aggregation function.
>
> > The authors mentioned that the Tweedie’s estimate in DPS causes high computational requirement [...]  exempt from the same limitation.
>
> We think that DPS converges worse because the overall framework assumes that the gradient of the terminal cost is accurate at every step, but it is actually approximated by Tweedie’s estimate. We think that our framework is able to deal with this approximation to the terminal cost better through the additional flexibility in the guidance. We propose rephrasing the corresponding section for the final manuscript to reflect this intuition.
>
> > I have reviewed the appendix [...] So far the qualitative comparison is not comprehensive.
>
> We include some more qualitative results on FFHQ [here](https://imgur.com/a/2cVdEff). In light of the new results on style guidance we will update the Appendix with more experimental details for these tasks, and additional qualitative results.
>
> > The most prominent concern I have about this paper is the lack of comparison with relevant literature [...] performance than the baselines selected by the authors.
>
> Thanks for your feedback and for suggesting other competitive baselines. Based on the reviewers' suggestion, we compare with two additional baselines, MPGD and RB-Modulation, on the non-linear deblur and 4x superresolution tasks for the FFHQ and ImageNet datasets. NDTM mostly outperforms these additional baselines on these tasks. Due to space constraints, please see our response to Reviewer UY5V (Limited Empirical Comparisons).
>
> > The authors choose  T=200,N=15  as the hyperparameters [...] Can the author explain why they can achieve this speedup?
>
> The configuration T=200, N=15 is used only for the blind inverse problem, which is inherently more challenging and benefits from additional optimization. However, for all other tasks, we use a more efficient configuration (See Table 8), carefully balancing performance and runtime. The runtimes reported in Table 9 correspond to the superresolution task, where T=50, N=5. This setup allows our method to be over 14x faster than DPS on that task.

---

> > ### Comment · Reviewer_113Q · 2025-04-02
> >
> > Thank you for your rebuttal! It has answered most of my concerns and I can see that the authors has put a lot of efforts into comparing with more prior works. Given that the authors have addressed my main concern, which is the lack of comprehensive comparison to baselines, I would increase my score to 3. However, I would still like to comment on several points from the authors' response:
> >
> > 1. **Linearity of the blind inverse problem task:** In DMPlug, whose setup is adopted in this paper according to line 325, the blind inverse problem experiment is about " about recovering a sharp image x (and kernel k) from y = k*x+n where * denotes the linear convolution and the spatially invariant blur kernel k is unknown" (page 8 in DMPlug). This makes this experiment linear. Please still consider modifying your claim regarding this matter.
> >
> > 2. **FID:** Thank you for your acknowledgment. My suggestion was simply to remove the claim about FID in your abstract since it is not an appropriate metric for your experiment setup, even though many prior works also make the same mistake.
> >
> > 3. **$\gamma$:** Thank you for your clarification. Another question: Is $\gamma$ a fixed value or a time-dependent value?
> >
> > Please answer my followup questions and I will raise my score!

---

> > > ### Author Response · Authors · 2025-04-02
> > >
> > > We again thank the reviewer for their feedback which helped make our work better. Please find our response to additional questions below:
> > >
> > > 1. **Re. linearity of the blind inverse problem**: Thanks for pointing this out. We will update our experimental setup to reflect these caveats.
> > >
> > > 2. **Re. FID**: Thanks for the suggestion. We will update our abstract in light of these arguments. Note that the claim needs to be updated anyway since we added new baselines for the non-linear deblur experiment.
> > >
> > > 3. **$\gamma$**: For simplicity, in this work, we assume $\gamma$ to be a fixed scalar (see line 172, right side in the main text). However, we can also impose a schedule on $\gamma$ (making it time-dependent). We will make this more explicit in Section 6 by highlighting it as an interesting direction for further exploration.
> > >
> > > Thanks again for considering to increase your score. We hope our response clarifies your concerns and we would be happy to answer further questions.

---

### Official Review · Reviewer_DxJn · 2025-03-17

**Overall Recommendation:** 3

**Summary:**

The paper proposes to optimize the guidance signal with three losses $C_{\text{score}}, C_{\text{control}} and C_{\text{terminal}$. After that, the guidance signal is utilized in the sampling process as normal guidance. The guidance signal is updated through the greedy scheme.

**Claims And Evidence:**

no problem

**Essential References Not Discussed:**

There are recent some advance in classifier-free guidance but does not have the discussion in the main paper.

[1] https://arxiv.org/abs/2404.07724

[2] https://arxiv.org/html/2404.13040v1

**Experimental Designs Or Analyses:**

Good

**Methods And Evaluation Criteria:**

1. The method might be too expensive. The running time cost should be provided

2. What is the value of N in Algorithm 1?

3. The evaluation does not have generation task for example label-condition generation, text2image generation. This hinders the evaluation of the work.

4. There are recent advances in classifier-free guidance but there is no discussion or comparison in the main paper.

[1] https://arxiv.org/abs/2404.07724

[2] https://arxiv.org/html/2404.13040v1

**Other Comments Or Suggestions:**

n/a

**Other Strengths And Weaknesses:**

n/a

**Questions For Authors:**

1. Please compare the running time of the algorithm with other CFG methods.

2. What is the value of N in Algorithm 1?

3. The evaluation does not have generation task for example label-condition generation, text2image generation. This hinders the assessment of the work. Please include these tasks in the paper.

4. The authors should include the analysis of the method when combining with other methods such as Interval Guidance, CFG++

**Relation To Broader Scientific Literature:**

Provide a new guidance method with optimization on guidance term before doing guidance

**Theoretical Claims:**

I checked the theoretical claims

---

> ### Author Rebuttal · Authors · 2025-04-01
>
> We thank the reviewer for their feedback. We address specific concerns as follows in the order of sections:
>
> > The method might be too expensive. The running time cost should be provided
>
> > Please compare the running time of the algorithm with other CFG methods.
>
> Runtime is often faster than the closest competitor. See the runtimes in Table 2 in the main text and Table 9 in Appendix C.2, where we list runtimes once per hyperparameter configuration. In detail, our method is about one order of magnitude faster than DPS, the closest competitor for superresolution and random inpainting. While our method is indeed slower than for example Red-diff or DDRM, it always improves the sample quality. Note that we did optimize all hyperparameters for sample quality for all methods. We agree that reducing the sampling time is a crucial improvement for guiding diffusion sampling, but this limitation applies to all high-quality competitors.
>
> > What is the value of N in Algorithm 1?
>
> We provide complete details of hyperparameters for our method and baselines in Tables 4,5 and 6 in the Appendix. To summarize, we set N=50 for linear and non-linear inverse problems and N=200 for blind inverse problems.
>
> > The evaluation does not have generation task for example label-condition generation, text2image generation. This hinders the evaluation of the work. [...]  Please include these tasks in the paper.
>
> Our framework indeed generalizes to non-linear tasks such as those pointed out by the reviewer. We demonstrate this by simply adapting the form of the terminal cost for Style Guidance generation and text-conditioned generation.
>
> **Style Guidance Generation**: Following the experimental setup in MPGD [He et al.], we evaluate our method on Style Guidance using Stable Diffusion 1.4. Please refer to our qualitative results at this URL: **[https://imgur.com/a/wEDqoaR](https://imgur.com/a/wEDqoaR)**. We further quantitatively evaluate our method on this task on 1000 (reference image, prompt) pairs and compare with FreeDOM [Yu et al.] and MPGD in the accompanying [**Table 1**](https://imgur.com/a/jTn0hQ2). Our quantitative and qualitative results suggest that our NDTM can be applied to this task and outperforms competing baselines on prompt (CLIP score) and style adherence (Style Score) while exhibiting better visual perceptual quality.
>
> **Text Guidance with CLIP**: Following the experimental setup in MPGD [1], we further present qualitative results on text-conditional generation using a pretrained diffusion model on the CelebA-HQ dataset, which can be accessed using this link: **[https://imgur.com/a/5jlnDLo](https://imgur.com/a/5jlnDLo)**. Similar to StyleGuidance, NDTM can be used to generate samples that adhere well to not only simple prompts (like Blond hair) but also more complex prompts such as “a person with blonde hair and a goatee”.
>
> Therefore, we emphasize that our method can be applied to more general guidance scenarios. For brevity, we will include additional qualitative results for these tasks with their experimental settings (like the form of the terminal cost) in a revised version of our paper. Note that we selected the best hyperparameters for the selected baselines, just like for all other experiments.
>
> We also urge the reviewer to check out our response Reviewer UY5V (see point Re. Limited Empirical Comparisons) which presents additional empirical comparisons with additional baselines on 4x super-resolution and non-linear deblur tasks
>
> > There are recent advances in classifier-free guidance but there is no discussion or comparison in the main paper. [see also] Essential References Not Discussed
>
> We thank the reviewer for the interesting pointers, which we will include in a dedicated section on classifier-free guidance in the related work section. Note that the two approaches are complementary: Classifier-free guidance is based on retraining a diffusion model using example training data, training-free guidance is based on a pretrained model and a cost function. We think an extensive evaluation of different paradigms is beyond the scope of this work.

---

### Decision · Program_Chairs · 2025-05-01

**Decision:**

Accept (poster)

**Comment:**

This paper proposes to model guided diffusion dynamics as a Markov chain and develops diffusion trajectory matching for training-free guidance. The paper originally received 2xWeakReject, 1xWeakAccept, and 1xAccept. The main concerns include running time, insufficient evaluations, unclear statements, etc. The authors have provided rebuttals and addressed most concerns of reviewers. Afterward, one reviewer raises the rating. Reviewer UY5V does not respond and keeps negative. However, the authors have clarified the concerns of Reviewer UY5V in the rebuttal. Considering the rebuttal and discussions from all reviewers, ACs recommend accepting this paper. The authors are suggested to carefully revise the paper and incorporate newly conducted experiments according to the comments and discussions.